# Role of protein synthesis and DNA methylation in the consolidation and maintenance of long-term memory in *Aplysia*

Kaycey Pearce[1], Diancai Cai[1], Adam C Roberts[1], David L Glanzman[1,2,3*]

[1]Department of Integrative Biology and Physiology, UCLA, Los Angeles, United States; [2]Department of Neurobiology, David Geffen School of Medicine at UCLA, Los Angeles, United States; [3]Integrative Center for Learning and Memory, Brain Research Institute, UCLA, Los Angeles, United States

**Abstract** Previously, we reported that long-term memory (LTM) in *Aplysia* can be reinstated by truncated (partial) training following its disruption by reconsolidation blockade and inhibition of PKM (Chen et al., 2014). Here, we report that LTM can be induced by partial training after disruption of original consolidation by protein synthesis inhibition (PSI) begun shortly after training. But when PSI occurs during training, partial training cannot subsequently establish LTM. Furthermore, we find that inhibition of DNA methyltransferase (DNMT), whether during training or shortly afterwards, blocks consolidation of LTM and prevents its subsequent induction by truncated training; moreover, later inhibition of DNMT eliminates consolidated LTM. Thus, the consolidation of LTM depends on two functionally distinct phases of protein synthesis: an early phase that appears to prime LTM; and a later phase whose successful completion is necessary for the normal expression of LTM. Both the consolidation and maintenance of LTM depend on DNA methylation.

**\*For correspondence:** glanzman@ ucla.edu

**Competing interests:** The authors declare that no competing interests exist.

## Introduction

Since the pioneering work of the Flexners (*Flexner et al., 1963*), Agranoff (*Agranoff and Klinger, 1964*) and their colleagues, it has been widely accepted that memory consolidation—the process by which a labile, short-term memory trace is transformed into a stable, long-term trace (*Lechner et al., 1999*; *Müller and Pilzecker, 1900*)—requires protein synthesis (*Davis and Squire, 1984*; *Goelet et al., 1986*; *Hernandez and Abel, 2008*). Moreover, the protein synthesis underlying memory consolidation has been observed to exhibit a temporal gradient: inhibition of protein synthesis during or shortly after training appears to be maximally disruptive of memory consolidation; significantly less amnesia is caused by protein synthesis inhibition (PSI) that commences approximately an hour or more after training (*Davis and Squire, 1984*).

In general, there has been little evidence for a functional distinction with respect to memory consolidation between protein synthesis that occurs during training—hereafter 'early' protein synthesis—and protein synthesis that occurs within the first hour or so after the end of training—hereafter 'late' protein synthesis. Thus, studies have classically observed significant disruptive effects on the consolidation of long-term memory (LTM) whether a protein synthesis inhibitor is applied either immediately prior to, or immediately after, training (e.g., *Agranoff et al., 1966*; *Barondes and Cohen, 1968*; *Flexner and Flexner, 1968*). Consequently, both early and late protein synthesis are commonly regarded as participating in a more-or-less unitary consolidative process. In particular, protein synthesis is believed to mediate critical late events in memory consolidation, including late

**eLife digest** The formation of long-term memory depends on new proteins being made in the brain. These new proteins are used partly to build the new connections among neurons that essentially store the memory, and must be made within a critical period of time. Experiments on animals have found that new proteins must be made during or shortly after training to form a stable memory; if protein synthesis is blocked during this period, the memory will not be stabilized (a process also known as memory consolidation).

Changes that alter the activity of genes in neurons also play essential roles in memory consolidation. One such change involves the attachment of a methyl group – a molecule that contains one carbon atom surrounded by three hydrogen atoms – to the DNA of a gene. This process, called DNA methylation, typically inhibits the activity of the gene.

Pearce et al. looked at how completely preventing protein synthesis and DNA methylation disrupted memory consolidation in a type of marine snail called *Aplysia*. Previously, researchers have exploited this animal's simple nervous system and behavior to discover basic biological mechanisms of memory that are common to all animals.

The snails were given training that increased the likelihood that they would reflexively withdraw part of their body (called the siphon) in response to touch. When Pearce et al. inhibited protein synthesis soon after training, the snails did not remember the training when tested 24 hours later, as expected. Further analysis showed, however, that a trace of the memory, referred to as the "priming trace", remained. Snails that had this priming trace could form a long-term memory after partial training, whereas untrained snails did not form memories after such partial training.

Inhibiting the synthesis of proteins during the original training blocked the priming trace, as did inhibiting DNA methylation during or after training. Moreover, inhibiting DNA methylation erased a previously established memory and prevented it from being reinstated by partial training.

Overall, the findings of Pearce et al. show that proteins produced in the brain by learning have multiple roles. In addition, both the consolidation and maintenance of long-term memory depend on one or more genes that otherwise suppress memory being inhibited via DNA methylation. Future work will now aim to identify the priming trace and the genes that suppress memory. Knowledge of the priming trace could lead to new treatments for memory-related disorders such as Alzheimer's disease. Furthermore, identifying genes that can suppress memory might allow us to reduce some of the harmful effects of traumatic experience.

gene transcription via the synthesis of transcription factors, such as the CCAAT/enhancer-binding protein (C/EBP), and the consequent synthesis of proteins involved in the construction of new synaptic connections (*Bailey et al., 2015*; *Kandel et al., 2014*).

One mechanism increasingly implicated in the consolidation of LTM is the epigenetic process of DNA methylation (*Levenson et al., 2006*; *Maddox et al., 2014*; *Miller et al., 2008*; *Monsey et al., 2011*; *Oliveira, 2016*; *Rajasethupathy et al., 2012*). However, the relationship between protein synthesis and DNA methylation in memory consolidation is unclear. Mechanistically, is protein synthesis upstream or downstream of DNA methylation during consolidation? DNA methylation is usually associated with gene silencing. If DNA methylation is required for the synthesis of necessary consolidative proteins, this would imply that a prerequisite for this synthesis is the silencing of one or more repressor genes. On the other hand, it is possible that activation of DNA methyltransferase (DNMT), the family of enzymes that catalyze the transfer of a methyl group to DNA, during memory consolidation itself depends on protein synthesis. Of course, these two possibilities are not mutually exclusive.

Here, we have examined the potentially distinctive roles of early and late protein synthesis in the consolidation of the LTM for behavioral sensitization in *Aplysia*. In addition, we have tested the effect on memory consolidation of both early and late inhibition of DNA methylation. We find that LTM can be induced by partial training, which is insufficient to induce LTM in naïve (untrained) animals, after the disruption of LTM by late, but not early, administration of a protein synthesis inhibitor. By contrast, both early and late inhibition of DNMT block LTM consolidation as indicated by the

preclusion of subsequent memory induction by partial training. These results point to a functional distinction between early and late protein synthesis in memory consolidation, and suggest a potential role for early protein synthesis in DNA methylation. Finally, we show that inhibition of DNMT disrupts not only the consolidation, but also the persistence, of LTM; thus, the maintenance of consolidated LTM requires ongoing DNA methylation.

## Results

### LTM can be induced by truncated sensitization training following amnesia produced by posttraining PSI

Animals were given training that induced long-term sensitization (LTS) of the siphon-withdrawal reflex (SWR); this training (hereafter 5X training) consisted of five bouts of tail shocks spaced 20 min apart (*Cai et al., 2011*, *2012*; *Chen et al., 2014*) (*Figure 1*). Control animals received no training. Then, ~15 min (range = 10–20 min) after training, trained animals received an intrahemocoelic injection of anisomycin, a protein synthesis inhibitor, or vehicle solution. Control animals received an injection of vehicle solution at the equivalent experimental time. At 24 h after training the duration of the SWR was tested in all the animals (24-h posttest), after which two of the groups—the control group (Control-Veh-3XTrained) and a group that had received the long-term training plus the posttraining injection of anisomycin (5XTrained-Aniso-3XTrained) were given additional sensitization training, which consisted of three bouts of tail shocks spaced 20 min apart. Previously, we found that this training (hereafter 3X training), which is insufficient to induce LTM in naïve animals, can successfully reinstate LTM following its disruption by inhibition of PKM Apl III—the *Aplysia* homolog of PKMζ (*Bougie et al., 2012*, *2009*)—or memory reconsolidation blockade (*Chen et al., 2014*). All the groups, including the two that received the 5X training but not the 3X training (5XTrained-Veh and 5XTrained-Aniso groups), were given another test at 48 h after training or at the equivalent experimental time (48-h posttest).

The group given 5X training followed by vehicle injection (5XTrained-Veh group) exhibited significant sensitization at 24 h after training compared with the two groups that received the 5X training followed by anisomycin injection (5XTrained-Aniso and 5XTrained-Aniso-3XTrained groups), and with the control group (Control-Veh-3XTrained). The subsequent 3X training produced LTS in the 5XTrained-Aniso-3XTrained group, as shown by the results for the 48-h posttest. (Notice that the 3X training did not induce LTS in animals that did not receive prior 5X training.) Thus, although posttraining PSI produced complete amnesia for sensitization at 24 h, it did not preclude the subsequent establishment of LTS by partial training.

Previous work (*Montarolo et al., 1986*) examined the effect of PSI at various times after training on serotonin (5HT)-dependent, long-term facilitation (LTF) of the sensorimotor synapse in dissociated cell culture, an in vitro homolog of LTS in *Aplysia* (*Kandel, 2001*). This work demonstrated that LTF, tested at 24 h after training, was disrupted by anisomycin applied during a 3-h period that extended from 1 h before the onset of spaced 5HT training through 0.5 h after training (*Montarolo et al., 1986*). (The 5HT training lasted 1.5 h.) By contrast, a 3-h period of anisomycin treatment beginning at either 0.5 h or 4 h after the end of 5HT training did not block LTF. Considering our behavioral results (above), we wished to determine whether exposure to a protein synthesis inhibitor immediately after 5HT training would disrupt LTF at 24 h and, if so, whether LTF could subsequently be induced by partial training. Accordingly, some sensorimotor cocultures were treated with anisomycin (10 µM, 2 h) immediately after training with five 5-min pulses of 5HT (100 µM; 5X5HT training), spaced at 15-min intervals (*Cai et al., 2008*; *Chen et al., 2014*) (*Figure 2A*). To test whether LTF could be induced by partial training following its potential disruption by posttraining anisomycin treatment, three spaced pulses of 5HT (3X5HT training) were used. The experiment included a group of cocultures (5X5HT group) that received the full 5HT training, but not the posttraining anisomycin, as well as a group (Control) that received neither the 5HT nor the posttraining anisomycin. Finally, there was a group (3X5HT) that received the three pulses of 5HT at 24 h, but not the initial 5X5HT training. (Cocultures not treated with a drug at a particular point in the experiment were treated with standard perfusion medium instead.) The application of anisomycin immediately after the 5X5HT training blocked LTF at 48 h (5X5HT-Aniso group; *Figure 2B,C*). However, three pulses of 5HT applied 24 h after the original 5HT training and subsequent PSI induced LTF at 48

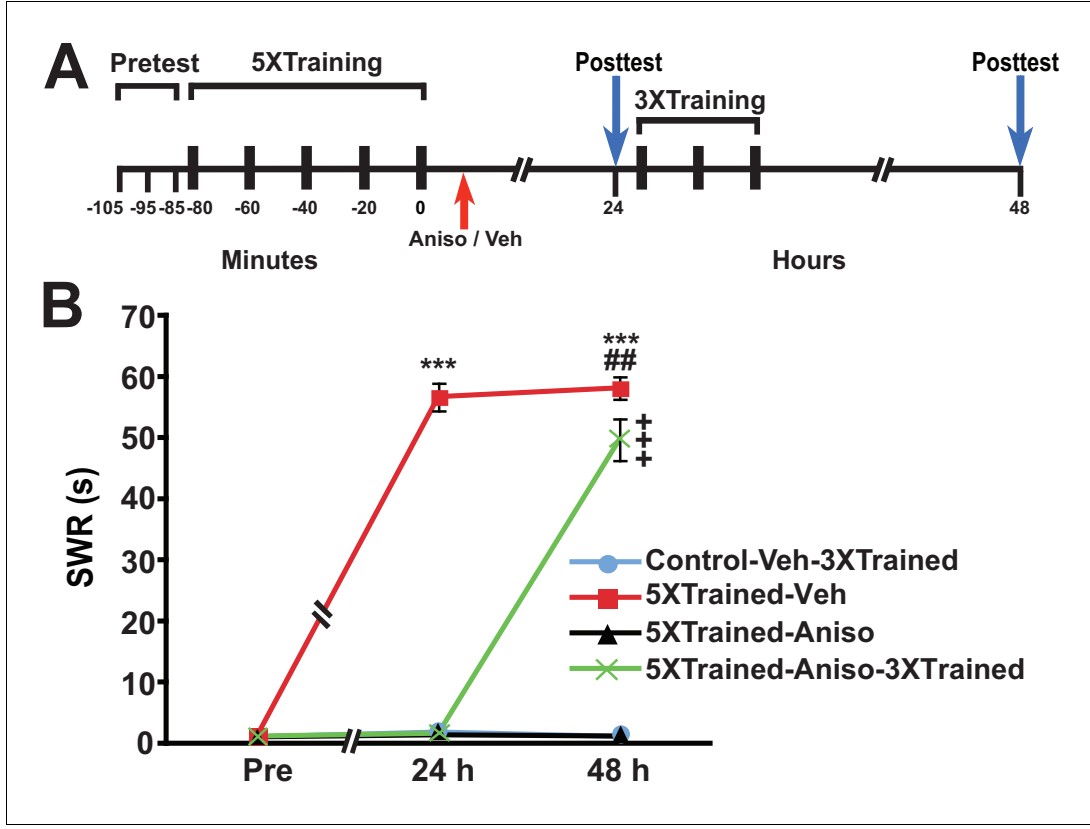

**Figure 1.** LTS can be established by truncated training following its disruption by posttraining PSI. (**A**) Experimental protocols. The times of the pretests, training, posttests, and drug/vehicle injections are shown relative to the end of the fifth bout of sensitization training (time = 0). The time of the intrahemocoelic injection of anisomycin or vehicle is indicated by the red arrow. After the 24-h posttest, animals in the Control-Veh-3XTrained and 5XTrained-Aniso-3XTrained groups received truncated sensitization training (3 bouts of tail shocks). (**B**) The mean duration of the SWR measured at 24 h and 48 h for the Control-Veh-3XTrained (*n* = 7), 5XTrained-Veh (*n* = 6), 5XTrained-Aniso (*n* = 5), and 5XTrained-Aniso-3XTrained (*n* = 6) groups. A repeated-measures ANOVA indicated that there was a significant group x time interaction ($F_{[6,40]}$ = 210.9, p < 0.0001). Subsequent planned comparisons indicated that the overall differences among the four groups were highly significant on all of the posttests (24 h, $F_{[3,20]}$ = 456.7, p < 0.0001; and 48 h, $F_{[3,20]}$ = 250.6, p < 0.0001). SNK posthoc tests revealed that the initial sensitization training produced significant LTS, as indicated by the increased mean duration of the SWR, in the 5XTrained-Veh group (56.7 ± 2.2 s) at 24 h compared with that in the Control-Veh-3XTrained (1.9 ± 0.9 s, p < 0.001). The mean duration of the SWR in the 5XTrained-Veh group at 24 h was also significantly longer than that in the 5XTrained-Aniso (1.4 ± 0.4 s, p < 0.001), and 5XTrained-Aniso-3XTrained groups (1.7 ± 0.7 s, p < 0.001). The differences among the Control-Veh-3XTrained, 5XTrained-Aniso and 5XTrained-Aniso-3XTrained groups were not significant at 24 h. The mean duration of the SWR in the 5XTrained-Veh group (58.2 ± 1.8 s) was still protracted at 48 h, and was significantly longer than that in the Control-Veh-3XTrained group (1.1 ± 0.2 s), as well as that in the 5XTrained-Aniso group (1.2 ± 0.2 s, p < 0.001 for both comparisons). LTS was induced in the 5XTrained-Aniso-3XTrained group by the three additional tail shocks applied after the 24-h posttest. The mean duration of the SWR in this group at 48 h was 49.7 ± 3.4 s, which was significantly longer than that for the Control-Veh-3XTrained group. In addition, the mean duration of the reflex in the 5XTrained-Aniso-3XTrained group was longer than that in 5XTrained-Aniso group, but still significantly shorter than that in the 5XTrained-Veh group at 48 h (p < 0.01). *Asterisks*, comparisons of the 5XTrained-Veh group with the Control-Veh-3XTrained group, the 5XTrained-Aniso group, and 5XTrained-Aniso-3XTrained group at 24 h; and comparisons of the 5XTrained-Veh group with the Control-Veh-3XTrained group and the 5XTrained-Aniso group at 48 h. *Pound signs*, comparison of the 5XTrained-Veh with the 5XTrained-Aniso-3XTrained group at 48 h. *Plus signs*, comparisons of the 5XTrained-Aniso-3XTrained group with the Control-Veh-3XTrained and 5XTrained-Aniso groups at 48 h. Here and in subsequent figures one symbol indicates p < 0.05; two symbols, p < 0.01; and three symbols, p < 0.001. Error bars in this and subsequent figures represent ± SEM.

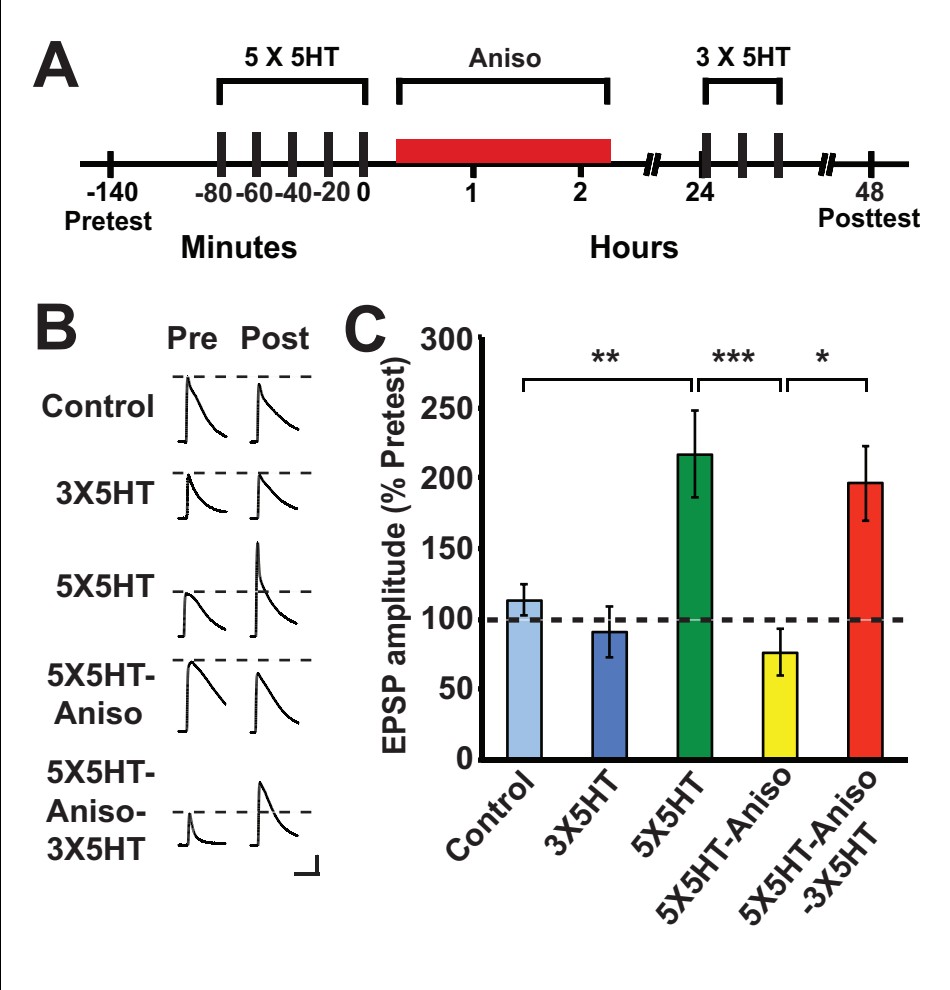

**Figure 2.** Partial training induces LTF following its disruption by PSI immediately after long-term training. (**A**) Experimental protocols. The initial training consisted of five 5-min pulses of 100 μM 5HT (5X5HT) spaced at 15-min intervals. Cocultures in the 5X5HT-Aniso and 5X5HT-Aniso-3X5HT groups were treated with anisomycin (10 μM, red bar) for 2 h immediately after the 5X5HT training. Three 5-min pulses of 5HT (100 μM; 3X5HT training) were given to cocultures in the 5X5HT-Aniso-3X5HT group at 24 h after 5X5HT training, as well as to cocultures in the 3X5HT group at the equivalent experimental time. (**B**) Sample EPSPs. Each pair of traces shows EPSPs recorded from the same coculture on the pretest and posttest. Scale bars: 10 mV, 100 ms. (**C**) Graph presenting the mean normalized EPSPs, measured at 48 h, for the five experimental groups: Control ($n = 13$), 3X5HT ($n = 12$), 5X5HT ($n = 16$), 5X5HT-Aniso ($n = 12$), and 5X5HT-Aniso-3X5HT ($n = 7$). A one-way ANOVA indicated that the overall differences among the five groups were highly significant ($F_{[4,55]} = 7.9$, $p < 0.0001$). SNK posthoc tests showed that the mean normalized EPSP in the 5X5HT group ($216.6\% \pm 30.6\%$) at 48 h was significantly larger than that in the Control ($113.1\% \pm 11.1\%$, $p < 0.01$), 3X5HT ($90.8\% \pm 17.9\%$, $p < 0.001$), and 5X5HT-Aniso ($76.1\% \pm 16.4\%$, $p < 0.001$) groups. The mean normalized EPSP in the 5X5HT-Aniso-3X5HT group ($196.5\% \pm 26.8\%$) was also significantly larger than that in the Control ($p < 0.05$), 3X5HT ($p < 0.05$), and 5X5HT-Aniso ($p < 0.05$) groups. None of the other differences among the groups was significant.

h (5X5HT-Aniso-3X5HT group). Notice that the mean EPSP in the 5X5HT-Aniso-3X5HT group was not significantly different from that in the 5X5HT group, which indicates that the supplemental partial training induced normal LTF. These cellular results accord with our behavioral finding that, although immediate posttraining PSI disrupts the consolidation of LTM, later abbreviated training can result in full LTM. *Montarolo et al. (1986)* observed significant LTF when cocultures were treated with anisomycin for 3 h, and even for 22 h, beginning 0.5 h after the end of 5X5HT training; the present results, taken together with those of Montarolo et al., indicate that

protein synthesis during a period of 30 min or so immediately following the 5X5HT training is critical for the normal consolidation of LTF in *Aplysia*.

## Amnesia produced by PSI during training cannot be subsequently reversed by partial training

*Castellucci et al. (1989)* found that PSI during the original period of sensitization training produced amnesia. We wished to determine whether LTM could be induced by partial training following its disruption by PSI during sensitization training. Accordingly, we performed an experiment using the identical protocol as that shown in *Figure 1*, except that the animals received an injection of either anisomycin or vehicle ~15 min before the onset of the 5X training (*Figure 3*). The 5X training induced LTM at the 24-h posttest in animals given the vehicle (Veh-5XTrained group), but not in animals given anisomycin (Aniso-5XTrained and Aniso-5XTrained-3XTrained groups) prior to training. Furthermore, 3X training did not produce LTM in animals that received the protein synthesis inhibitor prior to 5X training, as indicated by the lack of sensitization in the Aniso-5XTrained-3XTrained group at 48 h. In contrast to posttraining PSI, therefore, PSI during training prevented induction of LTM by the supplemental truncated training.

## Inhibition of DNA methyltransferase, whether during or shortly after training, causes irreversible amnesia

Why should PSI during training be so devastating for the consolidation of LTM? An intriguing possibility is that PSI during training obstructs DNA methylation required for memory consolidation. To test this possibility, we performed experiments in which the DNA methyltransferase (DNMT) inhibitor RG108 was injected into animals just before the onset of 5X training (*Figure 4*). DNMT inhibition during 5X training resulted in amnesia at 24 h and 48 h posttraining (comparison of the Veh-5XTrained group with the RG-5XTrained group); furthermore, subsequent 3X training did not induce LTM, as shown by the 48-h data (comparisons of the RG-5XTrained-3XTrained group with the Veh-5XTrained and Veh-Control-3XTrained groups).

Next we examined whether DNMT inhibition that commenced after long-term training caused amnesia and, if so, whether subsequent abbreviated training resulted in LTM. Accordingly, we performed an experiment like the previous one, except that RG108 was injected into some animals 10–20 min after, rather than before, 5X training (*Figure 5*). Posttraining DNMT inhibition, like pretraining DNMT inhibition, blocked the consolidation of LTM, as shown by the absence of LTS at 24 h and 48 h (comparisons of the 5XTrained-Veh group with the Control-Veh-3XTrained and 5XTrained-RG groups). In addition, supplemental 3X training following posttraining DNMT inhibition did not induce LTM (comparisons of the 5XTrained-RG-3XTrained group with the 5XTrained-Veh and Control-Veh-3XTrained groups).

## Inhibition of DNA methyltransferase eliminates consolidated LTM

The failure of partial training to induce LTM following posttraining RG108 treatment, coupled with its ability to establish LTM following posttraining PSI, suggests that DNA methylation is a prerequisite for the establishment of the occult priming trace accessed by partial training after posttraining PSI. To examine this possibility, we gave animals long-term behavioral sensitization training and then administered the DNMT inhibitor 24 h after training (*Figure 6*). All animals that received the 5X training exhibited LTS at 24 h (5XTrained-Veh, 5XTrained-RG and 5XTrained-RG-3XTrained groups), as indicated by the comparison with the control group (Control-Veh-3XTrained). LTS was also present at 48 h in animals that received the original long-term training and an injection of the vehicle after the 24-h test (5XTrained-Veh group), but not in animals that received the 5X training and an injection of RG108 at 24 h (5XTrained-RG and 5XTrained-RG-3XTrained groups). Moreover, truncated training did not restore LTM in the RG108-treated animals (comparison between the Control-Veh-3XTrained and 5XTrained-RG-3XTrained groups). An additional experiment using a different DNMT inhibitor—5-azacytidine (5-aza)—yielded identical results (*Figure 6—figure supplement 1*).

Because the animals were tested at 24 h, which was just before the application of the DNMT inhibitor, it might be argued that the subsequent disruption of LTM was mediated by blockade of memory reconsolidation (*Cai et al., 2012*; *Nader, 2015*), rather than by the interruption of ongoing gene silencing per se. To examine this possibility, we performed an experiment like that shown in

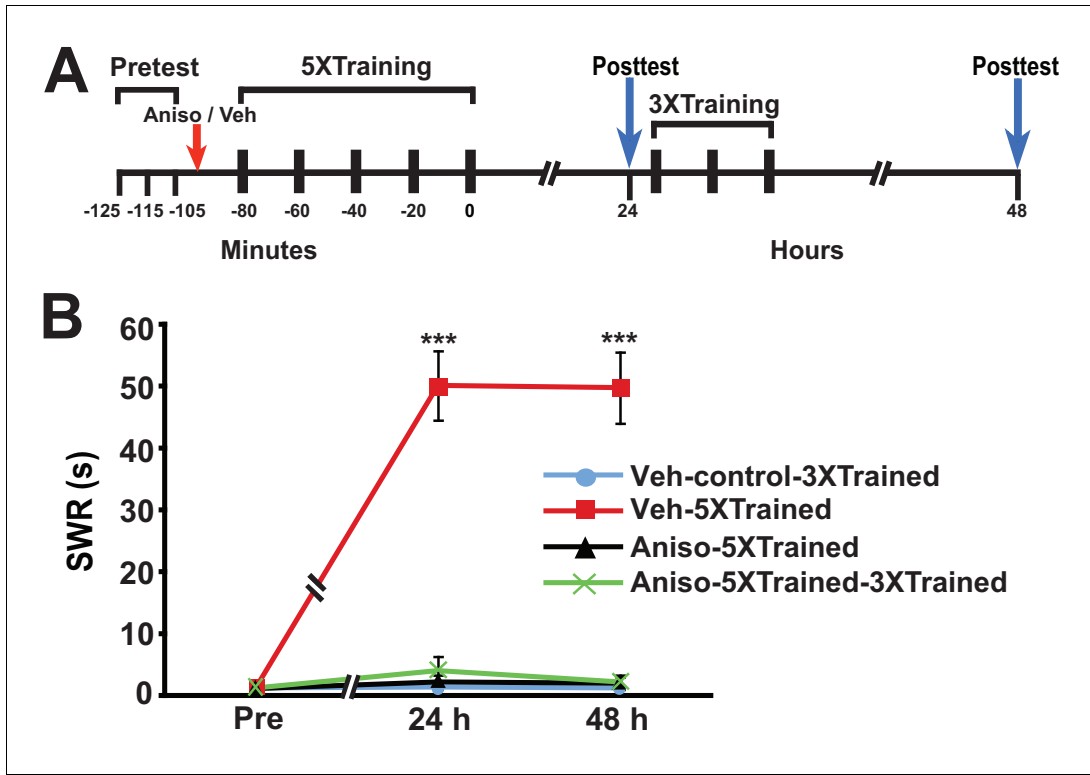

**Figure 3.** LTS cannot be induced by partial training when PSI occurs during the original (5X) sensitization training. (A) Experimental protocols. The times at which the pretests, training, posttests, and drug/vehicle injections occurred are shown relative to the end of the last training session. The red arrow indicates when either anisomycin or vehicle was injected into the hemocoel. (B) The mean duration of the SWR measured at 24 h and 48 h for the Veh-Control-3XTrained (n = 5), Veh-5XTrained (n = 8), Aniso-5XTrained (n = 6), and Aniso-5XTrained-3XTrained (n = 6) groups. A repeated-measures ANOVA showed a significant group x time interaction ($F_{[6,42]}$ = 40.9, p < 0.0001). Planned comparisons indicated that the group differences were highly significant for both 24-h ($F_{[3,21]}$ = 43.9, p < 0.0001) and 48-h ($F_{[3,21]}$ = 45.4, p < 0.0001) posttests. SNK posthoc tests revealed that the 5X training produced sensitization of the SWR in the Veh-5XTrained group (mean duration = 50.1 ± 5.6 s) at 24 h compared with the results for the Veh-Control-3XTrained group (mean duration of the SWR = 1.4 ± 0.4 s, p < 0.001). In addition, the SWR in the Veh-5XTrained group was significantly longer than that in the Aniso-5XTrained group (2.2 ± 1.2 s, p < 0.001) and the Aniso-5XTrained-3XTrained group (4.0 ± 2.3 s, p < 0.001). The differences among the Veh-Control-3XTrained, Aniso-5XTrained, and Aniso-5XTrained-3XTrained groups were not significant at 24 h. The SWR in the Veh-5XTrained group (mean duration = 49.8 ± 5.8 s) remained sensitized at 48 h, as indicated by the comparison with the reflex in the Veh-Control-3XTrained group (mean duration = 1.2 ± 0.2 s). The SWR was also significantly prolonged in the Veh-5XTrained group compared with that in the Aniso-5XTrained group (mean duration = 2.0 ± 1.0 s, p < 0.001 for both comparisons). The three tail shocks applied after the 24-h posttest did not establish LTS in the Aniso-5XTrained-3XTrained group. The mean duration of the SWR in this group at 48 h was 2.2 ± 1.2 s, which was not significantly different from that in the Veh-Control-3XTrained and Aniso-5XTrained groups. The SWR of the Veh-5XTrained group at 48 h was significantly longer than that in the Aniso-5XTrained-3XTrained group (p < 0.001). *Asterisks,* comparisons of the Veh-5XTrained group with the Veh-Control-3XTrained group, the Aniso-5XTrained group, and the Aniso-5XTrained-3XTrained group.

*Figure 6*, but omitted the 24-h test of the withdrawal reflex (*Figure 7*). Application of the DNMT inhibitor 5-aza a little more than 24 h after the 5X training eliminated LTM, as indicated by the lack of sensitization at 48 h in the 5XTrained-5aza and 5XTrained-5aza-3XTrained groups. In addition, treatment with 5-aza precluded the subsequent establishment of LTM by partial training (5XTrained-5aza-3XTrained group). Thus, the disruption of consolidated LTM by DNMT inhibition after training cannot be attributed to reconsolidation blockade.

A potential explanation for the devastating effect of DNMT inhibition on LTM at 24 h is that the memory for LTS is not fully consolidated by this time. To test whether inhibition of DNA methylation

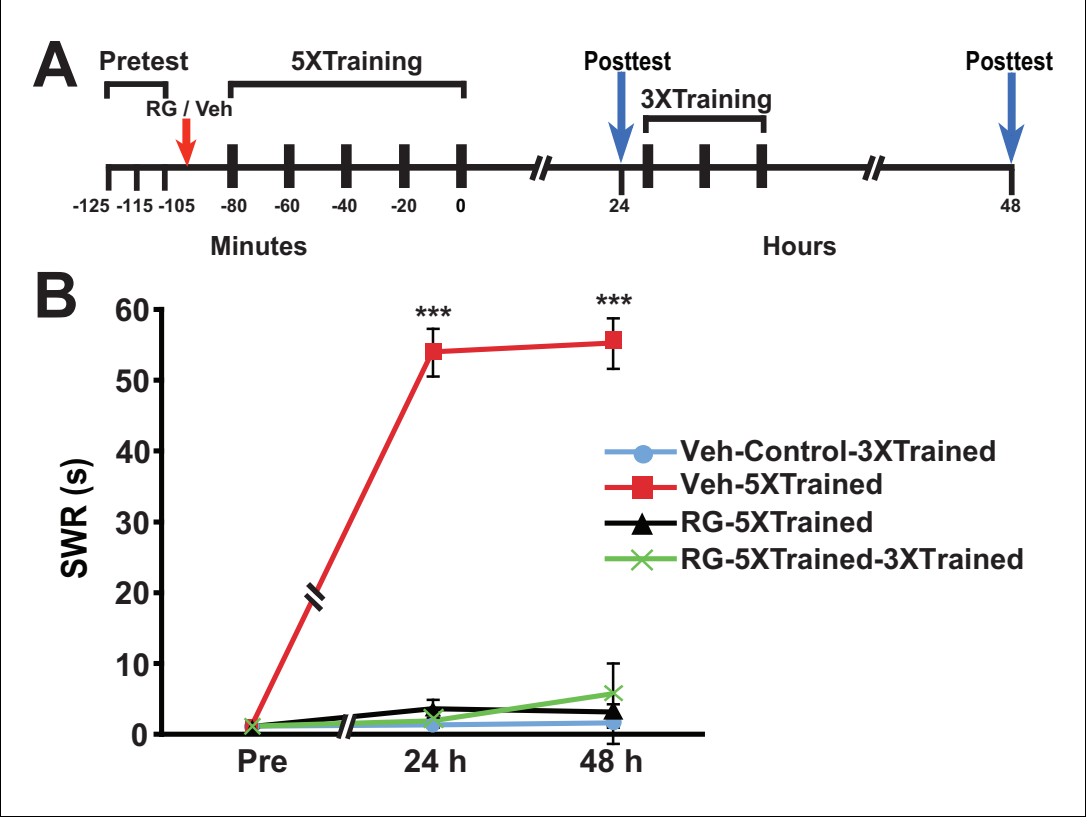

**Figure 4.** DNMT inhibition during the original (5X) sensitization training precludes the ability of subsequent partial training to induce LTS. (A) Experimental protocol. The times of occurrence of the pretests, training, posttests, and drug/vehicle injections are shown relative to the end of the last training session. Either RG108 or vehicle was injected into the hemocoel at the time indicated by the red arrow. (B) The mean duration of the SWR measured at 24 h and 48 h for the Veh-Control-3XTrained ($n = 7$), Veh-5XTrained ($n = 7$), RG-5XTrained ($n = 8$), and RG-5XTrained-3XTrained ($n = 7$) groups. A repeated-measures ANOVA indicated that there was a significant group x time interaction ($F_{[6,50]} = 73.6$, $p < 0.0001$). Subsequent planned comparisons showed that the overall differences among the four groups for the 24-h and 48-h posttests were highly significant (24 h, $F_{[3,25]} = 197.9$, $p < 0.0001$; and 48 h, $F_{[3,25]} = 82.8$, $p < 0.0001$). As revealed by SNK posthoc tests, the SWR exhibited sensitization at 24 h in the Veh-5XTrained group (mean duration = $54.0 \pm 3.4$ s) compared with that in the Veh-Control-3XTrained group (mean duration = $1.3 \pm 0.3$ s, $p < 0.001$). The differences in duration of the SWR at 24 h among the Veh-Control-3XTrained, RG-5XTrained ($3.6 \pm 1.3$ s), and RG-5XTrained-3XTrained ($1.9 \pm 0.6$ s) groups were not significant. Sensitization of the SWR was maintained in the Veh-5XTrained group (mean duration of the reflex = $55.3 \pm 3.6$ s) at 48 h, as shown by the comparison with the Veh-Control-3XTrained group (mean duration of the reflex = $1.6 \pm 0.6$ s, $p < 0.001$). There were no significant differences among the Veh-Control-3XTrained, RG-5XTrained (mean duration of the SWR = $3.1 \pm 1.2$ s), and RG-5XTrained-3XTrained (mean duration of the SWR = $5.7 \pm 4.4$ s) groups at 48 h, indicating that the three additional bouts of tail shocks given to the latter group after the 24-h posttest failed to induce LTS. *Asterisks,* comparisons of the Veh-5XTrained group with the Veh-Control-3XTrained group, the RG-5XTrained group, and the RG-5XTrained-3XTrained group.

would eliminate LTM at a later posttraining time, we performed experiments in which RG108 was administered 5 d after training. Here, animals were given the standard 5X training, tested for LTS 5 d later, and then given an injection of either RG108 or vehicle solution (LATE treatment condition) (*Figure 8A*). All animals were retested at 6 d, after which some animals were given 3X training. Finally, all animals were tested once more at 7 d. The 5X training produced LTS that persisted for 7 d after training (5XTrained-Veh$_{LATE}$ group vs. Control-Veh$_{LATE}$-3XTrained group) (*Figure 8B*). DNMT inhibition at 5 d after 5X training eliminated LTS (comparisons between the 5XTrained-RG$_{LATE}$ and 5XTrained-Veh$_{LATE}$ groups, and between the 5XTrained-RG$_{LATE}$ and Control-Veh$_{LATE}$-3XTrained groups), and partial retraining at 6 d failed to reinstate it (comparisons between

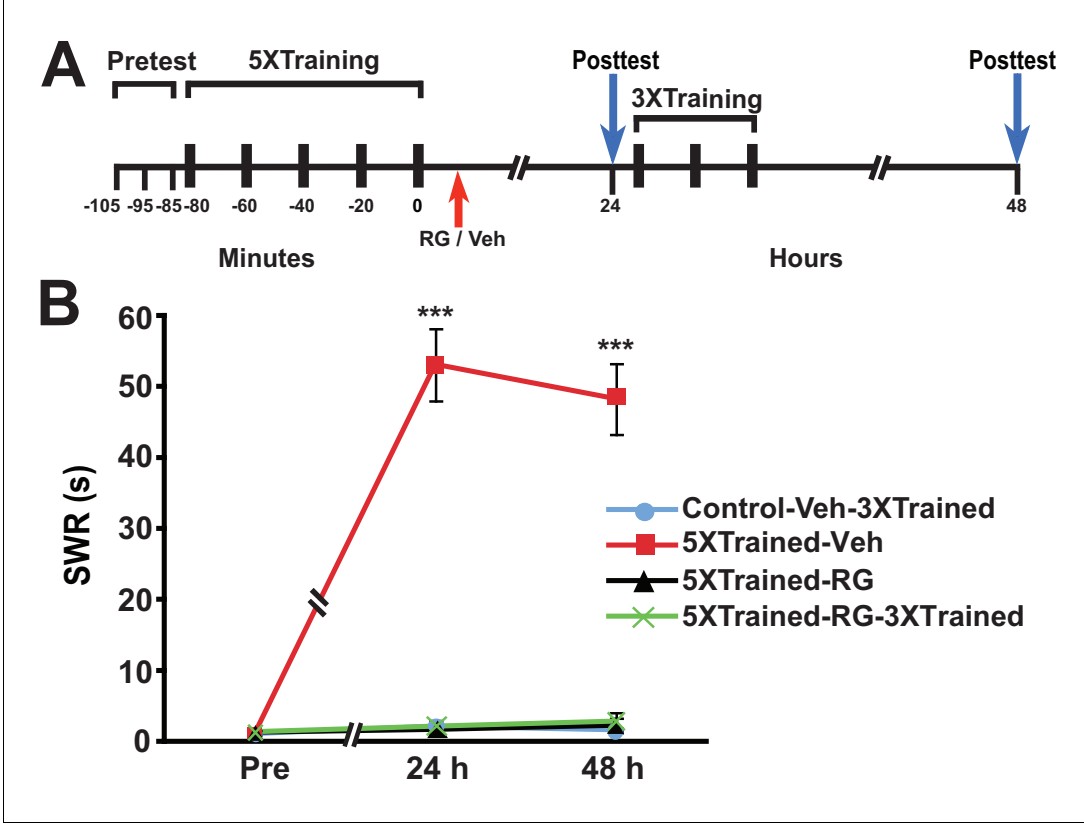

**Figure 5.** Posttraining inhibition of DNMT precludes later induction of LTS by partial training. (**A**) Experimental protocol. The times at which the pretests, training, posttests, and drug/vehicle injections occurred are shown relative to the end of the last training session. The time of the intrahemocoelic injection of either RG108 or vehicle is indicated by the red arrow. After the 24-h posttest, animals in the Control-Veh-3XTrained and 5XTrained-RG-3XTrained groups received 3X sensitization training. (**B**) The mean duration of the SWR measured at 24 h and 48 h for the Control-Veh-3XTrained ($n = 8$), 5XTrained-Veh ($n = 8$), 5XTrained-RG ($n = 7$), and 5XTrained-RG-3XTrained ($n = 6$) groups. A repeated-measures ANOVA showed that the group x time interaction was significant ($F_{[6,50]} = 64.7$, $p < 0.0001$). The overall differences among the four groups for the 24-h and 48-h posttests were highly significant, as indicated by a one-way ANOVA (24 h, $F_{[3,25]} = 82.6$, $p < 0.0001$; and 48 h, $F_{[3,25]} = 69.2$, $p < 0.0001$). SNK posthoc tests revealed significantly greater sensitization in the 5XTrained-Veh group at 24 h (mean duration of the SWR = 53.1 ± 5.1 s) than in the Control-Veh-3XTrained group (mean duration of the SWR = 2.1 ± 0.9 s, $p < 0.001$). The differences among the 5XTrained-RG (mean duration of the SWR = 1.7 ± 0.7 s), 5XTrained-RG-3XTrained (mean duration of the SWR = 2.2 ± 0.7 s), and Control-Veh-3XTrained groups at 24 h were not significant. Sensitization persisted in the 5XTrained-Veh group at 48 h (mean duration of the SWR = 48.3 ± 5.0 s) compared with the Control-Veh-3XTrained group (mean duration of the SWR = 1.6 ± 0.3 s, $p < 0.001$). The failure of the 3X training to induce sensitization in the 5XTrained-RG-3XTrained group was shown by the lack of significant differences between this group (mean duration of the SWR = 2.8 ± 1.2 s) and the Control-Veh-3XTrained group at 48 h. There was also no significant difference between the mean duration of the reflex in the 5XTrained-RG-3XTrained group and that in the 5XTrained-RG (2.3 ± 1.0 s) group at 48 h. *Asterisks,* comparisons of the 5XTrained-Veh group with the Control-Veh-3XTrained group, the 5XTrained-RG group, and the 5XTrained-RG-3XTrained group.

the 5XTrained-RG$_{LATE}$-3XTrained and 5XTrained-Veh$_{LATE}$ groups, and between the 5XTrained-RG$_{LATE}$-3XTrained and Control-Veh$_{LATE}$-3XTrained groups).

Another argument against the notion that consolidated LTM can be erased by DNMT inhibition is that DNMT inhibitors may injure the animals, or degrade their health or responsiveness in other ways. As a test for nonspecific, health-related effects of RG108, we assessed whether LTS could be reinduced in animals by 5X training following the elimination of LTS by administration of the DNMT inhibitor. We performed an experiment like that shown in *Figure 6*, except that animals given

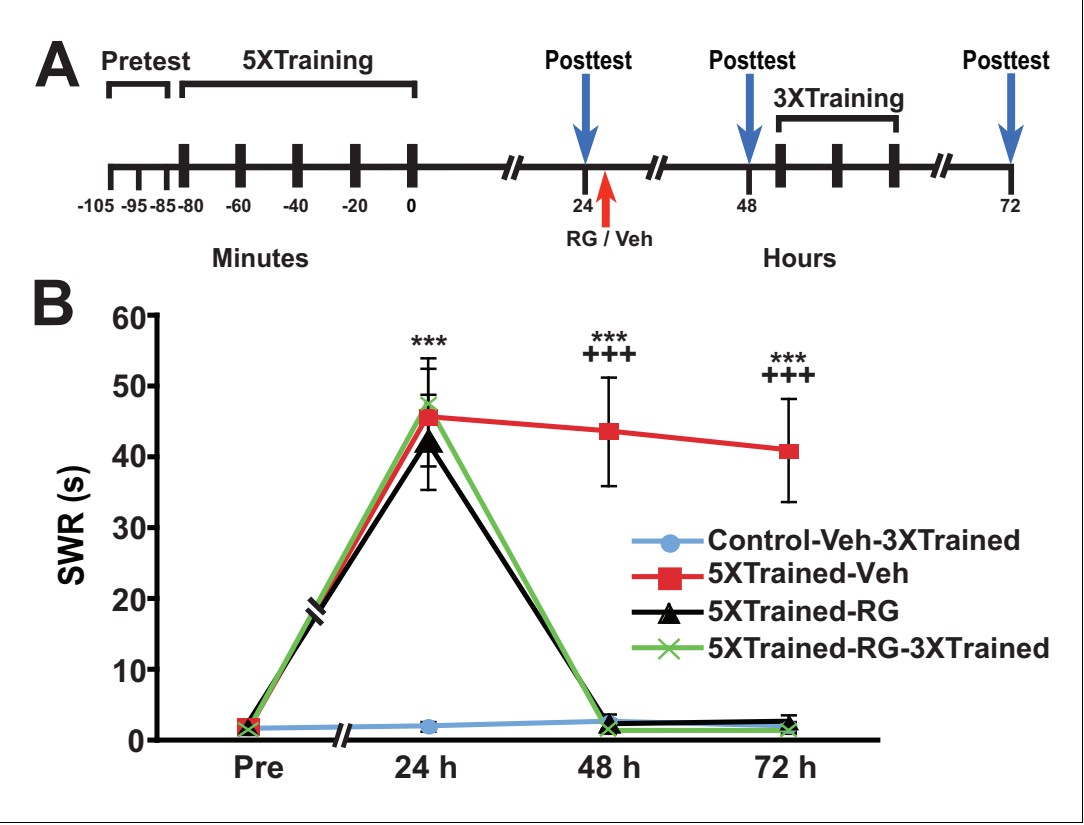

**Figure 6.** Inhibition of DNMT with RG108 eliminates established LTS in *Aplysia*. (**A**) Experimental protocol. The occurrences of the pretests, training, posttests, and drug/vehicle injections are shown relative to the end of the last training session. Either RG108 or vehicle was injected into the animals at the time indicated by the red arrow. After the 48-h posttest, animals in the Control-Veh-3XTrained and 5XTrained-RG-3XTrained groups received 3 bouts of sensitization training. (**B**) RG108 treatment at 24 h after training abolished LTS. There were four experimental groups: Control-Veh-3XTrained group (*n* = 6), 5XTrained-Veh group (*n* = 6), 5XTrained-RG group (*n* = 6), and 5XTrained-RG-3XTrained group (*n* = 6). A repeated-measures ANOVA disclosed a significant group x time interaction ($F_{[9,60]}$ = 22.9, p < 0.0001). Subsequent planned comparisons showed that the overall differences among the four groups for the 24-, 48- and 72-h posttests were highly significant (24 h, $F_{[3,20]}$ = 13.8, p < 0.0001; 48 h, $F_{[3,20]}$ = 28.6, p < 0.0001; and 72 h, $F_{[3,20]}$ = 27.9, p < 0.0001). Animals in all three groups trained with five bouts of tail shocks exhibited significant sensitization at 24 h, as indicated by SNK posthoc tests. Thus, the mean SWR was longer in the 5XTrained-Veh (45.7 ± 6.9 s), 5XTrained-RG (42.2 ± 6.7 s), and 5XTrained-RG-3XTrained (47.5 ± 6.5 s) groups than that in the Control-Veh-3XTrained group (2.0 ± 0.7 s; p < 0.001 for each comparison). However, although the 5XTrained-Veh group exhibited significant sensitization on both the 48-h (mean SWR = 43.7 ± 7.6 s) and 72-h (mean SWR = 41.0 ± 7.3 s) posttests, sensitization was absent in both groups of RG108-treated animals after 24 h. Posthoc tests revealed no significant differences for any of the comparisons between the Control-Veh-3XTrained group and the 5XTrained-RG group, or the 5XTrained-RG-3XTrained group, on the posttests after 24 h. Therefore, inhibiting DNMT with RG108 24 h after training erased established LTS. There was no evidence of spontaneous recovery of sensitization over the 48-h period after RG108 injection; furthermore, three additional bouts of training failed to reinstate LTS. *Asterisks,* comparisons of the 5XTrained-Veh, 5XTrained-RG, and 5XTrained-RG-3XTrained groups with the Control-Veh-3XTrained group at 24 h; and comparison of the 5XTrained-Veh group with the Control-Veh-3XTrained group at 48 h and 72 h. *Plus signs,* comparisons of the 5XTrained-Veh group with the 5XTrained-RG and 5XTrained-RG-3XTrained groups at 48 h and 72 h.

The following figure supplement is available for figure 6:

**Figure supplement 1.** Inhibition of DNMT with 5-azadeoxycytidine (5-aza) also eliminates established LTS in *Aplysia*.

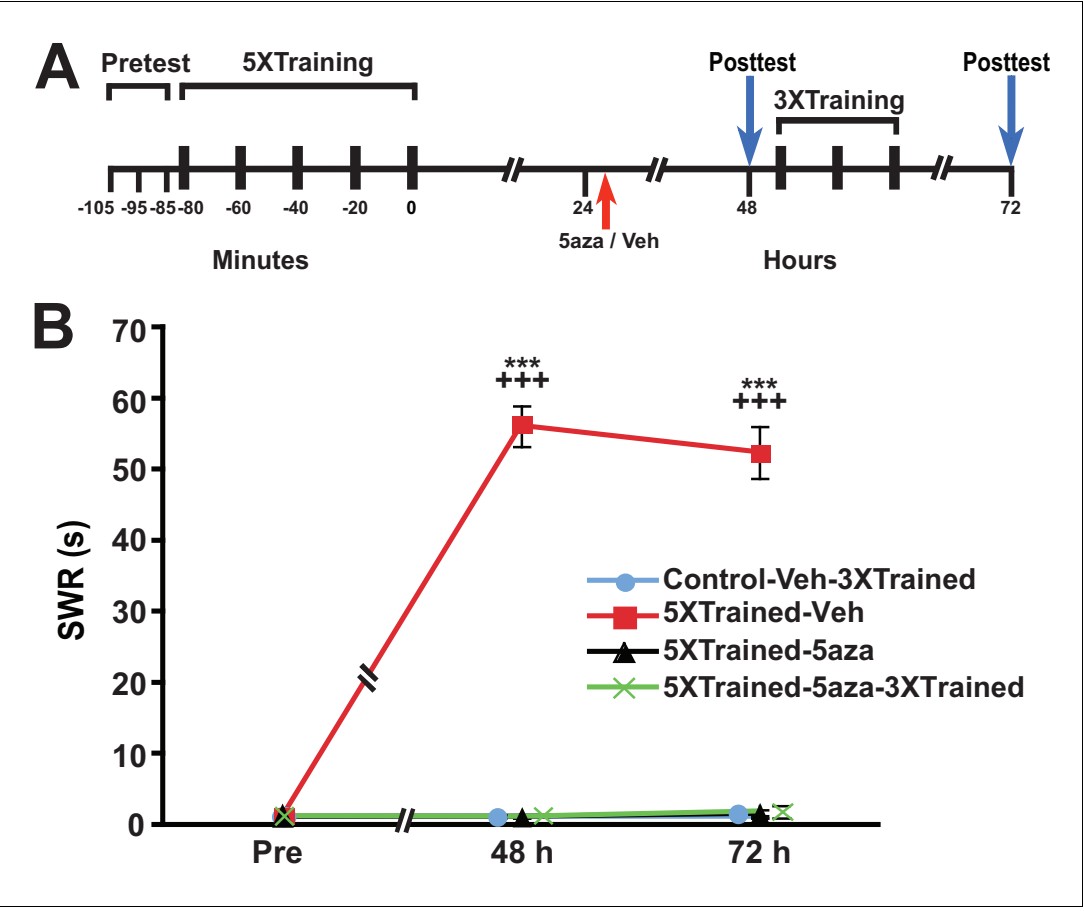

**Figure 7.** Disruption of established LTS with inhibition of DNMT is not a reconsolidation-related phenomenon. (**A**) Experimental protocol. The times at which the pretests, training, posttests, and drug/vehicle injections occurred are shown relative to the end of the last training session. The intrahemocoelic injection of either drug or vehicle is indicated by the red arrow. The animals did not receive a 24 h test prior to the drug/vehicle injection. After the 48-h posttest, animals in the Control-Veh-3XTrained and 5XTrained-5aza-3XTrained groups received 3 bouts of sensitization training (3X training). (**B**) 5-aza injection abolished established LTS in the absence of a posttest at 24 h. Four experimental groups were included: Control-Veh-3XTrained group ($n = 7$), 5XTrained-Veh group ($n = 7$), 5XTrained-5aza group ($n = 7$), and 5XTrained-5aza-3XTrained group ($n = 8$). A repeated-measures ANOVA indicated that there was a significant group x time interaction ($F_{[6,50]} = 105.8$, $p < 0.0001$). Subsequent planned one-way ANOVAs showed that the overall differences among the four groups at both 48 h and 72 h were highly significant (48 h, $F_{[3,25]} = 385.4$, $p < 0.0001$; and 72 h, $F_{[3,25]} = 183.3$, $p < 0.0001$). SNK posthoc tests revealed that the SWR in the 5XTrained-Veh group was significantly sensitized at both 48 h (mean = 56.1 ± 2.9 s) and 72 h (mean = 52.4 ± 3.7 s) compared with that in the Control-Veh-3XTrained group ($p < 0.001$ for each comparison). Furthermore, the mean duration of the SWR in the 5XTrained-Veh group was significantly longer than that in the 5XTrained-5aza (1.1 ± 0.1 s at 48 h, $p < 0.001$; and 1.6 ± 0.6 s at 72 h, $p < 0.001$) and 5XTrained-5aza-3XTrained (1.1 ± 0.1 s at 48 h, $p < 0.001$; and 1.9 ± 0.9 s at 72 h, $p < 0.001$) groups. The Control-Veh-3XTrained, 5XTrained-5aza, and 5XTrained-5aza-3XTrained groups did not differ significantly at either 48 h or 72 h. Thus, the erasure of established LTS by inhibition of DNMT (**Figure 6**) did not require elicitation of the SWR immediately preceding the drug injection. *Asterisks,* comparisons of the 5XTrained-Veh group with the Control-Veh-3XTrained, 5XTrained-5aza, and 5XTrained-5aza-3XTrained groups at 48 h and 72 h.

RG108 at 24 h received 5 bouts of training (that is, full LTS training) at 48 h rather than 3 bouts of training (partial training) (**Figure 9A**). As before, LTS was absent 24 h after administration of RG108 (comparisons between the 5XTrained-Veh group and the 5XTrained-RG and 5XTrained-RG-5XTrained groups at 48 h); nonetheless, 5X training after the 48-h test successfully reinduced LTS (comparisons between the 5XTrained-RG-5XTrained group and the 5XTrained-Veh and the

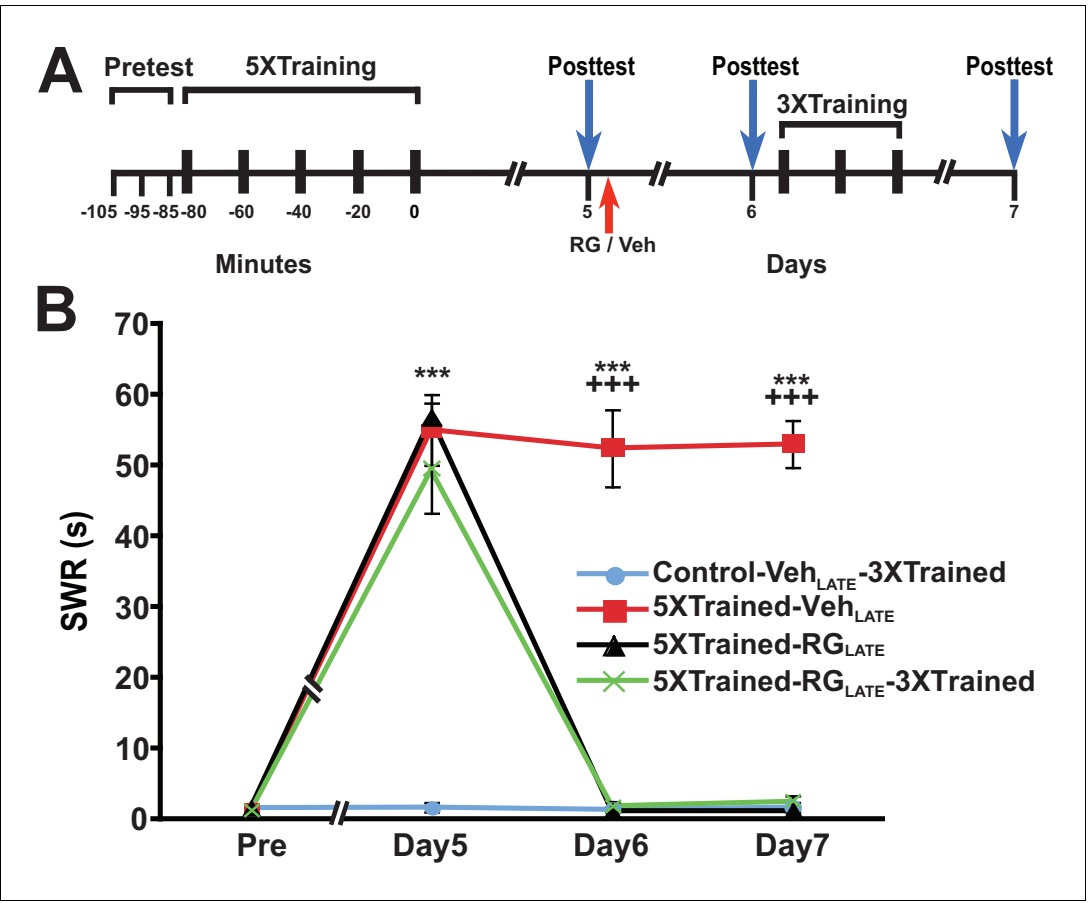

**Figure 8.** RG108 treatment 5 days after training abolishes LTS in *Aplysia*. (**A**) Experimental protocol. The occurrences of the pretests, training, posttests, and drug/vehicle injections are shown relative to the end of the last training session. The red arrow indicates the time of the intrahemocoelic injection of RG108 or vehicle. After the day six posttest, animals in some groups received partial sensitization training (3 bouts of tail shocks). (**B**) RG108 injection at day five after training (LATE treatment) erased LTS. There were four experimental groups: Control-Veh$_{LATE}$-3XTrained group ($n = 6$), 5XTrained-Veh$_{LATE}$ group ($n = 5$), 5XTrained-RG$_{LATE}$ group ($n = 6$), and 5XTrained-RG$_{LATE}$-3XTrained group ($n = 6$). A repeated-measures ANOVA indicated that there was a significant group x time interaction ($F_{[9,57]} = 66.3$, $p < 0.0001$). Subsequent planned comparisons showed that the overall differences among the four groups on days 5, 6 and 7 were highly significant (day 5, $F_{[3,19]} = 43.7$, $p < 0.0001$; day 6, $F_{[3,19]} = 105.1$, $p < 0.0001$; and day 7, $F_{[3,19]} = 252.5$, $p < 0.0001$). There was significant sensitization at day five prior to RG108/vehicle injection in the 5XTrained-Veh$_{LATE}$ (mean duration of the SWR = $55.0 \pm 5.0$ s), 5XTrained-RG$_{LATE}$ (mean duration of the SWR = $56.7 \pm 2.1$ s), and 5XTrained-RG$_{LATE}$-3XTrained (mean duration of the SWR = $49.3 \pm 6.1$ s) groups compared with the Control-Veh$_{LATE}$-3XTrained group (mean duration of the SWR = $1.7 \pm 0.7$ s) ($p < 0.001$ for each comparison). The 5XTrained-Veh$_{LATE}$ group also exhibited robust sensitization on day 6 (mean duration of the SWR = $52.4 \pm 5.5$ s) and 7 (mean duration of the SWR = $53.0 \pm 3.3$ s) compared with the Control-Veh$_{LATE}$-3XTrained group. Sensitization was absent in the 5XTrained-RG$_{LATE}$ group on day 6 and 7; thus, there was no spontaneous recovery of LTS during the 48-h period after the application of RG108. Three bouts of training shortly after the day six posttest did not restore LTS in the 5XTrained-RG$_{LATE}$-3XTrained group the next day. In particular, the mean duration of the SWR in the 5XTrained-RG$_{LATE}$-3XTrained group ($2.5 \pm 0.8$ s) on day 7 was not significantly different from that in the Control-Veh$_{LATE}$-3XTrained group, and was significantly shorter than that in the 5XTrained-Veh$_{LATE}$ group ($p < 0.001$). *Asterisks*, comparisons of the 5XTrained-Veh$_{LATE}$, 5XTrained-RG$_{LATE}$, and 5XTrained-RG$_{LATE}$-3XTrained groups with the Control-Veh$_{LATE}$-3XTrained group on day 5; and comparison of the 5XTrained-Veh$_{LATE}$ group with the Control-Veh$_{LATE}$-3XTrained group on days 6 and 7. *Plus signs*, comparisons of the 5XTrained-Veh$_{LATE}$ group with the 5XTrained-RG$_{LATE}$ and 5XTrained-RG$_{LATE}$-3XTrained groups on days 6 and 7.

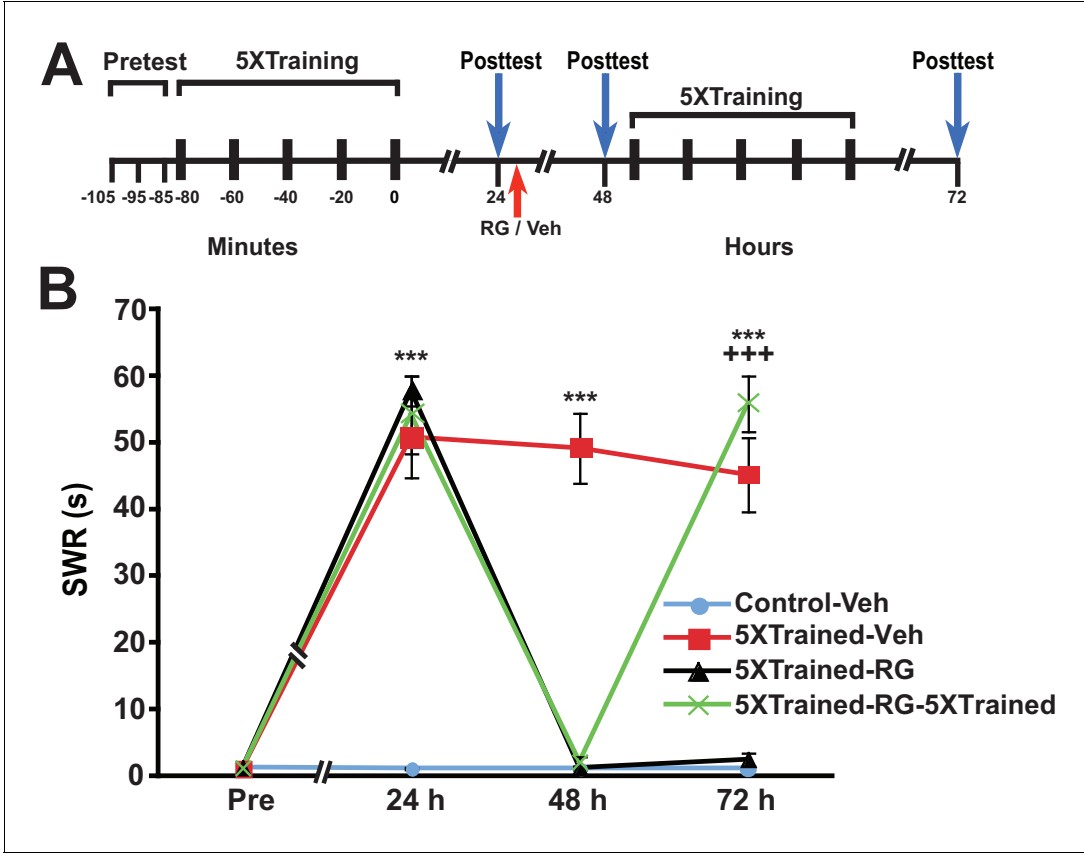

**Figure 9.** Animals can relearn following elimination of LTS by DNMT inhibition. (**A**) Experimental protocol. The times at which the pretests, training, posttests, and drug/vehicle injections occurred are shown relative to the end of the last training session. The time of the intrahemocoelic injection of either RG108 or vehicle is indicated by the red arrow. After the 48-h posttest, animals in the 5XTrained-RG-5XTrained group received a second round of full sensitization training (five bouts of electrical tail shocks). (**B**) Sensitization retraining produced LTS in animals following erasure of LTM by RG108. There were four experimental groups: Control-Veh group (*n* = 6), 5XTrained-Veh group (*n* = 6), 5XTrained-RG group (*n* = 4), and 5XTrained-RG-5XTrained group (*n* = 6). A repeated-measures ANOVA indicated that the group x time interaction was significant ($F_{[9,54]}$ = 61.4, p < 0.0001). Subsequent one-way ANOVAs indicated that the overall differences among the four groups at 24 h, 48 h and 72 h were highly significant (24 h, $F_{[3,18]}$ = 33.3, p < 0.0001; 48 h, $F_{[3,18]}$ = 70.7, p < 0.0001; and 72 h, $F_{[3,18]}$ = 54.9, p < 0.0001). SNK posthoc tests performed on the 24-h data revealed that the initial training produced significant sensitization in the 5XTrained-Veh group (mean duration of the SWR = 50.8 ± 6.1 s), 5XTrained-RG group (mean duration of the SWR = 57.8 ± 2.3 s), and 5XTrained-RG-5XTrained group (mean duration of the SWR = 54.2 ± 5.8 s) compared with Control-Veh group (mean duration of the SWR = 1.2 ± 0.2 s; p < 0.001 for each comparison). The SWR of 5XTrained-Veh group also exhibited sensitization at 48 h (mean duration = 49.2 ± 5.2 s) and 72 h (mean duration = 45.2 ± 5.6 s) compared with the Control-Veh group (48 h, mean duration = 1.2 ± 0.2 s, p < 0.001; and 72 h, mean duration = 1.2 ± 0.2 s, p < 0.001). Sensitization memory was significantly disrupted at 48 h in both the 5XTrained-RG (mean SWR = 1.3 ± 0.3 s) and 5XTrained-RG-5XTrained (mean SWR = 2.2 ± 0.7 s) groups by the RG108 injection immediately after the 24-h posttest (p > 0.05 for the comparisons with the Control-Veh group). Retraining after the 48-h posttest reestablished full LTM. The mean duration of the SWR in the 5XTrained-RG-5XTrained group at 72 h (55.8 ± 4.2 s) was significantly greater than that for the Control-Veh group (mean duration = 1.2 ± 0.2 s, p < 0.001), as well as for the 5XTrained-RG group at 72 h (2.5 ± 1.0 s, p < 0.001). *Asterisks,* comparisons of the 5XTrained-Veh, 5XTrained-RG, and 5XTrained-RG-5XTrained groups with the Control-Veh group at 24 h; comparisons of the 5XTrained-Veh group with the Control-Veh, 5XTrained-RG, and 5XTrained-RG-5XTrained groups at 48 h; and comparison of the 5XTrained-Veh group with the Control-Veh and 5XTrained-RG groups at 72 h. *Plus signs,* comparison of the 5XTrained-RG-5XTrained group with the 5XTrained-RG group at 72 h.

5XTrained-RG groups at 72 h) (*Figure 9B*). Thus, the apparent elimination of LTM following treatment with RG108 cannot be ascribed to a deleterious effect of the drug on the health of the animals. Taken together, our results strongly argue that the maintenance of LTM in *Aplysia* requires ongoing DNA methylation.

## Discussion

We have shown that protein synthesis during and shortly after sensitization training is essential for the normal consolidation of LTM in *Aplysia*. Our results therefore confirm previous results obtained in *Aplysia* by *Montarolo et al. (1986)* and *Castellucci et al. (1989)*, as well in vertebrates by many groups (reviewed in *Davis and Squire, 1984*; *Hernandez and Abel, 2008*). In addition, however, we have significantly extended prior findings regarding protein synthesis and memory consolidation through our demonstration that LTM can be induced by supplemental partial training following its disruption by PSI shortly after the original (full) LTS training, but not following PSI during the original LTS training. Thus, the present results reveal a novel functional distinction between the mnemonic role of protein synthesis during training and that of protein synthesis shortly after training. An early study in *Aplysia* indicated that bath-applied anisomycin (18 μM) produces rapid (≤15 min), nearly complete (95–99%) inhibition of protein synthesis, as measured by the incorporation of leucine into proteins in identified central neurons (*Schwartz et al., 1971*). Because the pretraining injections of anisomycin in our study were made 10–20 min prior to the onset of training, and because the duration of the 5X training in our study was ~80 min, the pretraining anisomycin treatment would be expected to produce >90% disruption of protein synthesis in the animals throughout much, if not all, of the training period. The posttraining injections of anisomycin in our study were made 10–20 min after the end of 5X training; if one assumes a maximum post-injection time of 15 min for the onset of significant PSI within the central nervous system (CNS) of the animals (*Schwartz et al., 1971*)—drugs injected into the hemocoel of *Aplysia* have ready access to the CNS due to the open circulatory system and lack of a blood-brain barrier in gastropod mollusks (*Abbott et al., 1986*)—then the posttraining injections of anisomycin should have begun to inhibit protein synthesis by >90% within 30 min after the end of 5X training. Our results indicate that proteins synthesized during training (early protein synthesis) play a special role in the consolidation of LTM. Specifically, early protein synthesis causes the generation of a priming component that allows LTM to be later established by partial training if it is disrupted by posttraining PSI (*Figures 1* and *2*). *Barzilai et al. (1989)* reported that a 1.5-h treatment with 5HT produces the rapid onset (<30 min) of the synthesis of several (unidentified) proteins; in addition, the synthesis of some of these proteins depends on gene transcription, suggesting that they represent immediate-early proteins. Thus, the priming component induced by early protein synthesis in our study may be the product of immediate-early gene transcription. In support of this idea, *Rajasethupathy et al. (2012)* found that exposure to five spaced pulses of 5HT down regulated the expression of the transcriptional repressor CREB2 in *Aplysia* sensory neurons; moreover, this downregulation depended on methylation of the *CREB2* gene, because it was blocked by the DNMT inhibitor RG108. CREB2 is regarded as a memory suppressor in *Aplysia* (*Abel et al., 1998*) (although see *Hu et al., 2015*); blockade of CREB2 activity (by means of a function-blocking antiserum) in *Aplysia* sensory neurons has been shown to facilitate the induction of LTF (*Bartsch et al., 1995*). The memory suppressive effect of CREB2 is due, at least in part, to the repression of CREB1 activation and the downstream expression of immediate-early genes, including the *C/EBP* gene (*Alberini et al., 1994*; *Alberini and Kandel, 2014*). Thus, the expression of CREB1-dependent immediate-early genes, enabled through 5HT-induced DNA methylation of *CREB2*, could form the priming memory component revealed by the present experiments. According to this idea, DNA methylation would be upstream from the early protein synthesis required for memory consolidation in *Aplysia*.

The blockade of protein synthesis shortly after (i.e., starting ~30 min after) training, albeit disruptive of LTM, as indicated by the absence of LTM at 24 h following posttraining PSI, nonetheless does not preclude the subsequent induction of LTM by partial training. (Note that our results appear to partly contradict those of *Montarolo et al. (1986)* who observed that anisomycin applied to sensori-motor cocultures starting at 30 min after 5X5HT training did not block LTF; however, the onset of the anisomycin application in our experiments was ~15 min earlier than in the Montarolo et al. study, which may explain the apparent discrepancy in results.) In support of the present findings, *Shobe et al. (2016)* have recently reported that consolidation of the LTM for sensitization is disrupted by posttraining application of the protein synthesis inhibitor emetine to a reduced preparation of *Aplysia*. Therefore, PSI after training does not disrupt the memory primer, which is induced by early protein synthesis (*Figure 3*). It seems likely, moreover, that it is the persistence of this

primer that underlies the ability of truncated training to reinstate LTM following its disruption by reconsolidation blockade or inhibition of PKM (*Chen et al., 2014*).

How can truncated training establish LTM following impairment of memory consolidation by posttraining PSI, and also reinstate consolidated LTM after its disruption by reconsolidation blockade or inhibition of PKM (*Chen et al., 2014*)? One possibility is that protein synthesis—in addition to signaling by one or more growth-related factors (*Hu et al., 2004*; *Kopec et al., 2015*; *Zhang et al., 1997*)—during long-term training results in persistent activation of mitogen-activated protein kinase (MAPK) (*Martin et al., 1997*; *Sharma et al., 2003b*), and it is the sustained MAPK activity that enables reinstatement of LTM by partial training. In support of this idea, sensitization training that is sufficient to induce LTM has recently been found to cause prolonged (1 h) posttraining activation of MAPK (*Shobe et al., 2016*). But posttraining PSI disrupts this persistent MAPK activity (*Shobe et al., 2016*), and we have shown that the memory priming signal is maintained despite posttraining blockade of translation (*Figures 1* and *2*). Although the priming signal cannot therefore be MAPK itself, it could be a signal downstream from activated MAPK. *Martin et al. (1997)* showed that during 5HT-induced LTF MAPK translocates to the nucleus of sensory neurons; this nuclear translocation is required for LTF. Possibly, the priming signal is a nuclear change downstream of MAPK. Epigenetic modifications are attractive candidates for the priming signal. Indeed, MAPK-dependent increases in the phosphorylation of histone H3 have been implicated in LTM in rats (*Chwang et al., 2006*) and snails (*Danilova et al., 2010*). Furthermore, we previously showed that inhibition of histone deacetylase (HDAC) permitted 3X sensitization training to induce LTM in naïve, untrained animals (*Chen et al., 2014*). Thus, persistent histone modifications induced by 5X training may provide the scaffolding necessary for later reconstruction of LTM by partial training during memory reinstatement. Besides MAPK-dependent epigenetic modifications, other candidates for the priming signal include small, non-coding RNAs, the expression of which is induced by 5HT training during LTF (*Rajasethupathy et al., 2012*). In future research we will seek to identify the memory priming signal.

Our demonstration that LTM can be fully established by abbreviated training after being disrupted by posttraining PSI echoes recent findings for contextual fear memory in mice by Tonegawa and colleagues (*Ryan et al., 2015*). These investigators used optogenetic stimulation of hippocampal neurons that had been active during fear conditioning to restore LTM after the induction of retrograde amnesia by posttraining treatment with anisomycin. Similar to our finding that LTF was absent at 24 h in sensorimotor cocultures after 5X5HT training followed by exposure to anisomycin (*Figure 2*), Ryan et al. observed an absence of learning-induced long-term potentiation (LTP) (*Herring and Nicoll, 2016*) in hippocampal slices from anisomycin-treated animals; this synaptic disruption correlated with retrograde amnesia. Despite the lack of persistent synaptic changes widely regarded as hallmarks of consolidated memory (*Bailey et al., 2015*; *Dudai et al., 2015*), some aspect of LTM nonetheless endured in the two studies.

*Ryan et al. (2015)* did not test whether LTM could be induced by optogenetic stimulation subsequent to PSI during fear conditioning, nor did they propose a specific storage mechanism for LTM. Our data point to DNA methylation as being essential for the consolidation of LTM. Prior work in mammals also supports a critical role for DNA methylation in memory consolidation (*Halder et al., 2016*; *Levenson et al., 2006*; *Miller et al., 2008*; *Miller and Sweatt, 2007*; *Monsey et al., 2011*; *Oliveira, 2016*). More relevantly, Kandel and colleagues reported that RG108 blocks the establishment of LTF in *Aplysia* sensorimotor cocultures, and that this effect is due, at least in part, to blockade of 5HT-induced methylation of the gene for the transcriptional repressor CREB2 (*Rajasethupathy et al., 2012*). The present results elaborate upon this idea; besides showing that the consolidation of LTM in *Aplysia* requires epigenetic suppression of one or more memory repressor processes, our results suggest the intriguing possibility that this suppression may depend on early protein synthesis. Although little is known at present regarding the potential role of protein synthesis in DNA modification, a report that activity-dependent induction of a specific gene, *Gadd45b*, is essential for DNA demethylation in the mammalian hippocampus (*Ma et al., 2009*) is consistent with this idea. Furthermore, in their study of the epigenetic regulation of memory consolidation in *Aplysia*, *Rajasethupathy et al. (2012)* found that DNA methylation of *CREB2* was regulated by a neuronally expressed Piwi protein. Perhaps the early protein synthesis mediating the consolidation of LTM involves the expression of Piwi in *Aplysia*. In the absence of direct evidence that protein synthesis triggers DNA methylation in *Aplysia*, however, it is at least as plausible that

the process of DNA methylation is upstream, rather than downstream, of early protein synthesis in the consolidation of LTM, as discussed above.

Given, as our results indicate, that the consolidation of LTM in *Aplysia* depends critically on the silencing of one or more genes whose protein products act to repress memory, what are potential candidates for this memory repressive function? An obvious candidate, of course, is *CREB2* (*Bartsch et al., 1995*; *Rajasethupathy et al., 2012*), but there are others. For example, phosphatases have been proposed to subserve memory repression in mammals. *Miller and Sweatt (2007)* found that infusion of a DNMT inhibitor into the hippocampus of rats immediately after contextual fear conditioning blocked the consolidation of fear memory as assessed 24 h later, and that this effect was due, in part, to DNA methylation of the gene for protein phosphatase 1 (PP1). Another phosphatase that may subserve memory repression, and whose gene may become silenced by DNA methylation during learning, is calcineurin (protein phosphatase 2B) (*Baumgärtel and Mansuy, 2012*). Calcineurin activity suppresses the induction of hippocampal LTP (*Winder et al., 1998*; *Winder and Sweatt, 2001*). Moreover, genetically overexpressing calcineurin in the brains of mice disrupts the consolidation of LTM (*Mansuy et al., 1998*), whereas genetically inhibiting calcineurin enhances hippocampal LTP and LTM in mice (*Malleret et al., 2001*). In addition, *Baumgärtel et al. (2008)* reported that calcineurin activity is inhibited in the amygdala during the consolidation of conditioned taste aversion (CTA) in mice, and that the level of calcineurin activity during learning determines the strength of the CTA memory. In *Aplysia* the effects of genetically inhibiting or overexpressing either PP1 or calcineurin on LTM have yet to be examined. However, both phosphatases modulate the CREB-mediated response to extracellular stimuli in *Aplysia* signaling pathways (*Hawkins et al., 2006*). Moreover, pharmacological inhibition of calcineurin has been shown to facilitate the induction of the LTM for sensitization in *Aplysia* (*Sharma et al., 2003a*).

Besides demonstrating a role for DNA methylation in memory consolidation, the present study shows that ongoing DNA methylation plays a crucial role in memory maintenance. This finding resonates with those from recent work in mammals (*Halder et al., 2016*; *Miller et al., 2010*; *Mizuno et al., 2012*), as well as in invertebrates (*Biergans et al., 2015*; *Lukowiak et al., 2014*). We do not know the identity of the gene or genes whose persistent methylation underlies memory maintenance in *Aplysia*. The gene for CREB2 is one possibility; but a recent study using sensorimotor cocultures found that a long-lasting *increase* in postsynaptic CREB2 expression mediated the maintenance of LTF beginning 48 h after the induction of synaptic plasticity (*Hu et al., 2015*), which is inconsistent with a role for the continuous silencing of the *CREB2* gene in the maintenance of sensitization memory. Interestingly, *Miller et al. (2010)* observed persistent DNA methylation of the gene for calcineurin in the anterior cingulate cortex of rats trained contextual fear conditioning. The notion that ongoing suppression of calcineurin activity via gene silencing mediates LTM maintenance in *Aplysia* is attractive in light of previous data implicating calcineurin activity in the inhibition of LTM (*Sharma et al., 2003a*); nonetheless, to date no direct evidence supports a role for calcineurin in the maintenance of memory in *Aplysia*.

A unique and fascinating aspect of the present results is that they enable a direct comparison of the effect on the persistence of a single, specific form of memory—LTS in *Aplysia*—of inhibiting DNA methylation with those for two other manipulations that have been purported to eliminate consolidated LTM, inhibition of PKMζ (*Sacktor, 2011*) and reconsolidation blockade (*Nader, 2015*). Previously, we showed that although inhibition of PKM Apl III, the *Aplysia* homolog of PKMζ (*Bougie et al., 2012, 2009*), and reconsolidation blockade both disrupt the consolidated LTM for behavioral sensitization in *Aplysia* (*Cai et al., 2011, 2012*), the memory can nonetheless be fully reinstated using truncated sensitization training (*Chen et al., 2014*). By contrast, as shown in the present study, the LTM for sensitization cannot be reinstated after its disruption by inhibition of DNMT (*Figures 6–9*). This finding indicates that the ongoing DNA methylation of one or more genes is a precondition for memory maintenance, and, moreover, that some essential priming component of LTM, one whose persistence enables LTM reinstatement, must be impervious to the disruptive effects of PKM inhibition and of reconsolidation blockade, but eliminable by inhibition of DNMT.

In summary, we have found that two functionally distinct periods of protein synthesis regulate the consolidation of LTM in *Aplysia*. The earlier period occurs during training (and, possibly, extends into the immediate posttraining period as well); it involves the production of a memory priming element. The later period starts within 30 min after training; proteins synthesized during this

posttraining period are required for the normal expression of LTM. Nonetheless, inhibition of the synthesis of these late proteins, albeit disruptive of LTM, does not impair the priming element, whose occult presence permits LTM to be established by supplemental partial training following its disruption by posttraining PSI. Finally, we have shown that both the consolidation and maintenance of LTM in *Aplysia* depend, in part, on gene silencing via DNA methylation. Future work will be required to determine the identity of the gene/s whose DNA methylation is required for the induction and persistence of LTM, and whether this methylation is triggered by early protein synthesis.

## Materials and methods

### Behavioral training and testing

Adult *Aplysia californica* (80–120 g) were supplied by Alacrity Marine Biological Services (Redondo Beach, CA, USA) and housed in cooled (12–14°C), aerated seawater. The behavioral training, testing, and drug injection methods were described in our previous study (*Chen et al., 2014*).

### Cell culture and electrophysiology

The cell culture, electrophysiological recording, 5HT training, and anisomycin treatment methods were like those previously described (*Cai et al., 2011*; *Chen et al., 2014*).

### Drug preparation

The protein synthesis inhibitor, anisomycin, was prepared for injection as was previously done (*Chen et al., 2014*). The DNA methyltransferase (DNMT) inhibitors RG108 (Sigma, St Louis, MO) and 5-azadeoxycytidine (5-aza; EMD Millipore, Billerica, MA) were dissolved in DMSO to a concentration of 25 mM to make a stock solution. To inhibit DNMT, a volume of 100 µl per 100 g of body weight of either RG108 or 5-aza was injected into the animals. The specific times at which the intrahemocoelic injections were made are indicated in the relevant figures.

### Statistical analysis

SPSS 22.0 (IBM, Armonk, NY) was used for the statistical analysis. For the analysis of the behavioral data, a repeated-measures two-way analysis of variance (ANOVA) was first performed on the overall data. Once the repeated-measures ANOVA showed a significant interaction, one-way ANOVAs were carried out on the separate test times, followed by Student-Newman-Keuls (SNK) posthoc tests for pairwise comparisons. For the synaptic data, the EPSP amplitude of the posttest was normalized to that of the pretest for the same coculture. The normalized data were expressed as means ± SEM. A one-way ANOVA was performed on the overall data, followed by Student-Newman-Keuls (SNK) posthoc tests for pairwise comparisons. All reported levels of significance represent two-tailed values unless otherwise indicated.

## Acknowledgements

This study was supported by research grants to DLG from the National Institute of Neurological Disorders and Stroke (NIH R01 NS029563), the National Institute of Mental Health (NIH R01 MH096120), and the National Science Foundation (IOS 1121690). We thank S Apichon, B Cheema, ME Kimbrough, V Kong, D Miresmaili, EJ Moc, A Rangchi, R Sumner, X Zhao and A Zobi for assistance with the behavioral training, and S Chen, A Bédécarrats, FB Krasne and WS Sossin for helpful comments on the manuscript.

## Additional information

### Funding

| Funder | Grant reference number | Author |
| --- | --- | --- |
| National Institute of Neurological Disorders and Stroke | NIH R01 NS029563 | David L Glanzman |
| National Institute of Mental | NIH R01 MH096120 | David L Glanzman |

| | | |
|---|---|---|
| | Health | |
| National Science Foundation | IOS 1121690 | David L Glanzman |

The funders had no role in study design, data collection and interpretation, or the decision to submit the work for publication.

## Author contributions

KP, Conceptualization, Data curation, Investigation, Methodology; DC, Conceptualization, Data curation, Formal analysis, Investigation, Writing—original draft, Writing—review and editing; ACR, Software, Formal analysis, Methodology; DLG, Conceptualization, Formal analysis, Supervision, Funding acquisition, Methodology, Writing—original draft, Project administration, Writing—review and editing

## Author ORCIDs

David L Glanzman, http://orcid.org/0000-0001-5479-0245

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
