## [Decision Letter]

Thank you for submitting your article "Effects of Pretraining and Posttraining Injections of a Protein Synthesis Inhibitor on Memory Consolidation in *Aplysia*" for consideration by *eLife*. Your article has been favorably evaluated by K VijayRaghavan (Senior Editor) and two reviewers, one of whom, Mani Ramaswami (Reviewer #1), is a member of our Board of Reviewing Editors, and another one is Thomas Abrams (Reviewer #2).

The reviewers have discussed the reviews with one another and the Reviewing Editor has drafted this decision to help you prepare a revised submission.

The authors use an interesting set of training protocols in combination with protein synthesis inhibition (PSI) in an attempt to dissect the contribution of molecular processes to memory consolidation vs. expression. They also introduce a DNA methyltransferases inhibitor to explore the contribution of DNA methylation to memory consolidation. The difference in the effects of protein synthesis inhibition and DNMT inhibition suggests an interesting explanation for recent provocative results from the authors' lab.

The manuscript describes strong and clear data in support of the following novel and original conclusions in the well-known and influential *Aplysia* sensorimotor circuit that mediates the siphon-withdrawal reflex (SWR).

A) They confirm that two temporally distinct periods of protein synthesis during and 15 minutes after "full" long-term sensitization (LTM) training (5X5HT) are necessary for 24-hour memory. However, here they show that early protein synthesis alone can lay down a long-lasting, normally invisible memory trace (previously termed "priming") that the authors reveal here through clever experimentation.

B) Namely, that in 5x5HT-treated preparations where late protein synthesis has been inhibited but early protein synthesis has occurred, an "abbreviated" 3X5HT training delivered 24 hours afterwards can induce stable LTM visible 24 hours after abbreviated training. This does not occur if early protein synthesis is inhibited or if abbreviated stimulation is applied on its own to a fresh preparation.

Less convincingly, the authors also suggest that early protein synthesis appears to have this long-lasting effect (enabling 3X5HT induced LTM) at least partially through methylation of DNA. The recovery of memory by abbreviated LTM training that is observed fails if DNA methyltransferases are inhibited, suggesting that ongoing DNA methylation may be the mechanism for a component of memory persistence after late PSI. This is an important conclusion that could be drawn from the present experiments, but somehow is not explicitly discussed or sufficiently advanced in the manuscript.

However, the main conclusion of the Research Advance is that there is a specific function for early protein synthesis in laying down a long-lasting component of the LTM memory trace (which the authors had previously agreed was appropriately termed as "priming'), even if these are not visible to normal behavioral experiments. This finding is interesting, novel and merits publication.

Major points that must be addressed:

Most critical, the language and presentation are confusing and (perhaps unsurprisingly) for many of the reasons that were already brought up in the original article on which this Advance is based. This needs to be clarified, partially in the same manner as for the original article. In addition, there are some simple experimental lacunae that need to be filled out to better establish or nuance the authors' conclusions.

1) The term consolidation should be almost systematically replaced by an expression that conveys the main message to the reader. "Consolidation," in contemporary parlance should surely be used to include all steps from initial induction to the stable expression of memory, particularly in this simple preparation where EPSP magnitude is a satisfactory surrogate for memory expression. The long-lasting cryptic effect of 5X5HT stimulation that persists after late PSI should be termed "priming" consistent with the original manuscript. The authors' observations can be understood and communicated simply by proposing that in neurons primed by 5XHT induced early-protein synthesis, subsequent events required for LTF can be induced by 3X5HT training. This simplified presentation/ discussion would take nothing away from the novelty of the observation. Few would have thought that a long-lasting nuclear mark would persist stably in neural circuits after LTM training, if the circuit itself has never encoded stable LTM.

To explain our position, we point out that most neuroscientists would agree that consolidation is not one event, but involves multiple phases. One of which would be a nuclear priming event in which some nuclear processes required for LTM consolidation are activated. Another phase is revealed by this work to require late protein synthesis. There are also several other examples of multiple stages consolidation in *Aplysia*. The MAPK pathway is recruited relatively late during training protocols, and may well still be contributing to memory formation for tens of minutes after the last block of sensitizing stimuli (i.e., tail shocks). Chain et al. (1999) found that inhibiting PKA presynaptically within several hours after a series of 5-HT applications almost completely blocked long-term facilitation tested at 24 hours; the dependence on PKA activity appeared to gradually decrease within the first 12 hours, which presumably represents one mechanism of consolidation.

The overly simplistic and – we suggest – incorrect dichotomy between memory consolidation and expression, also ignores the molecular evidence that memory maintenance is an active process. The authors' previous work implicates activity of truncated PKC (PKM) in memory maintenance at times as late as 1-3 days after training.

2) On a related note, the term "retrieval" is also used in a very confusing fashion and this needs to be addressed systemically in from the Introduction through the Discussion. The authors observe that early PSI blocks EPSP potentiation (which is the mechanism of memory encoding). Early PSI does blocks memory expression/ LTF consolidation. Retrieval is never tested. The paper doesn't address the issue of memory retrieval. Rather it addresses two stages of memory encoding and the existence of a stable and rate-limiting initial process of (probably nuclear) priming.

In further explanation of our position, if consolidation of memory were truly complete by 30 min after training, then a protein synthesis inhibitor that blocked retrieval or expression would have no persistent effect once it had been eliminated from the CNS. If the protein synthesis inhibitor anisomycin selectively affected expression, this effect should require the presence of anisomycin at the time of the posttest. In *Aplysia*, the effects of anisomycin disappear within approximately 3 hours, as demonstrated by Montarolo et al. (1986). Because posttests were conducted at 24 hours and anisomycin effects should have ended at least 20 hours earlier, there should be no remaining effects on retrieval at the time of the posttest. Furthermore, the efficacy of post training anisomycin in blocking memory tested at 24 hours is highly dependent on the precise time of anisomycin administration. Montarolo et al. transiently applied anisomycin only shortly after the time point chosen in the present experiments; they found that memory retention was largely protein synthesis independent 30 min after a series of 5-HT exposures, and completely protein synthesis independent 4 hours after the end of the 5-HT. If protein synthesis inhibition impaired retrieval, the effect should be the same for applications at these several time points shortly after training. (The Montarolo et al. paper should be cited when the post training anisomycin experiment is first introduced.) Even blocking protein synthesis for 22 hours but beginning 30 min after a series of 5-HT exposures, did not dramatically block long-term facilitation measured at 24 hours. Thus, these earlier published data from *Aplysia* would strongly suggest that inhibition of protein synthesis 15 minutes after the end of 1.5 hour training protocol does not selectively impair expression of memory without affecting consolidation. Interestingly, although the authors cite Montarolo et al., they don't consider this earlier evidence that consolidation is still progressing at the post training time point that they apply anisomycin.

"Our results support a retrieval-related account of retrograde amnesia." As emphasized above, there is really no evidence presented for this conclusion.

"The subsequent 3X training restored LTM in the 5XTrained-Aniso-3XTrained group, as shown by the results for the 48 h posttest..… Thus, post training PSI did not block either the induction or the consolidation of LTM, but merely its expression at 24 h." This interpretation permeates the text, which leads readers, and possibly the authors, to ignore more plausible explanations for their observations, as argued above.

3) It is important to further assess whether test if the persistence of "priming," is reflected by the persistence of early DNA methylation, or if DNA demethylation and remethylation occur for long periods after training (as suggested by David Sweatt's 2010 analysis of fear conditioning).

3a) How is LTF and "priming" in response to full LTM training are affected by DNMT inhibitors applied 4 hours after training? The 4-hour time point is suggested because Montarolo and colleagues (1986) have shown after 4 hours, 24 hr LTF induced by 5X5HT treatment is independent of new protein synthesis.

Again, in explanation: The work here shows that whereas inhibition of protein synthesis could be "overcome" or rescued by an abbreviated additional training protocol (three bouts of tail shocks delivered at 24 hours), inhibition of DNA methyltransferases produced a memory deficit that could not be rescued by the same abbreviated additional training. Does this suggest memory maintenance requires persistent DNMT activity, as found by Miller, Sweatt and colleagues, 2010? Or is DNA methylation the signal that persists after protein synthesis inhibition (and blockade of reconsolidation) and that primes memory formation when there is additional abbreviated training? (These possibilities are not mutually exclusive.) Rajasethupathy et al. found that piRNAs are upregulated by 5-HT application during long-term facilitation, and these piRNAs promote methylation of the promoter for the inhibitor CREB, CREB2, suppressing its expression. The authors' data suggest that maintenance of DNA methylation does not require protein synthesis, but rather is perhaps upstream from protein synthesis, which is consistent with the Rajasethupathy results. Nevertheless, the authors suggest that perhaps "PSI during training obstructs DNA methylation required for memory consolidation," citing Rajasethupathy et al. This proposal should be revised, as it contradicts the authors' data and also the Rajasethupathy et al. findings. Importantly, the Rajasethupathy et al. results are consistent with the understanding that memory formation is regulated by a balance between repressors and activators of memory consolidation; the "priming effect" on memory formation by subsequent abbreviated training protocol observed here could involve suppression of expression of CREB2 due to DNA methylation, a possibility that the authors should discuss.

3b) Similarly, if ongoing DNA methylation provides the molecular component of memory/priming that persists after protein synthesis inhibition, or after blockade of reconsolidation, one would predict that late treatment with a DNMT inhibitor, e.g. at 19 or 20 hours, might block the efficacy of abbreviated training in producing full LTM detectable at 48 hours. The authors should discuss this point.

The authors should also present additional results with RG108, to include in Figure 4, indicating how DNMT inhibition of "primed" preparations shortly before abbreviated training at the 24-hour time point affects the ability of the abbreviated training to restore LTM. Is there a reinforcement of a pre-set DNA methylation pattern required for LTM restoration? If LTM restoration is independent of ongoing DNA methylation, then the authors would have their strongest evidence for DNA methylation being a mechanism for priming. If ongoing DNA methylation is required LTM restoration, this would also be very interesting.

4) The authors should perform one experiment and discuss candidate molecular cellular processes controlled by abbreviated (3X5HT) training in naive and primed preparations. In primed preparations (24 hours after PSI delivered 15 minutes after 5X5HT training), what does protein-synthesis inhibition before and after 3X5HT stimulation do to LTM consolidation?

In explanation, the abbreviated training protocol with three bouts of tail shock is likely to initiate some of the same processes that contribute to memory formation as the full five-bout protocol. It is important to acknowledge that the abbreviated protocol may be sufficient for formation of memory because some residual components of memory persist after treatment with the protein synthesis inhibitor, such as DNA methylation and to provide at least one or two simple experiments to restrict the possible range of processes recruited by 3X5HT training.

[Editors' note: further revisions were requested prior to acceptance, as described below.]

Thank you for resubmitting your article "Role of Protein Synthesis and DNA Methylation in the Consolidation and Maintenance of Long-Term Memory in *Aplysia*" for consideration by *eLife*. Your article has been reviewed by one peer reviewer (Thomas Abrams, Reviewer #2), and the evaluation has been overseen by Mani Ramaswami as Reviewing Editor and K VijayRaghavan as the Senior Editor.

The reviewers have discussed the reviews with one another and the Reviewing Editor has drafted this decision to help you prepare a revised submission.

Summary:

In long-term sensitization of the defensive withdrawal reflex of *Aplysia*, established memory can be interrupted with either blockade of persistently active PKC or blockade of reconsolidation. The authors showed that a trace of memory persists that can prime the restoration of memory, so that it occurs with only modest supplemental training. In this manuscript, the authors show that protein synthesis block 15 minutes after LTF training blocks memory consolidation but also similarly leaves a trace that primes subsequent memory restoration. They explore whether and now this persistent component of memory that survives the retraction of newly formed synapses is localized to the nucleus. The originally submitted Research Advance manuscript was quite strong, but with the addition of new experiments that test the effect of DNA methyl transferase inhibitors applied after memory has been fully consolidated, this manuscript now is greatly improved and enhanced and should much higher impact. The intriguing conclusion is that ongoing DNA methylation is essential for memory maintenance even days after training. One particularly compelling experiment tested the efficacy of late, 5x training after earlier exposure to the DNMT inhibitor to confirm that learning is still possible.

Essential revisions:

1) Is the recruitment of DNA methyltransferases that contribute to long-term memory consolidation and maintenance downstream of protein synthesis? This is an unresolved issue, and its discussion in the manuscript needs to be improved and clarified.

1a) The authors state: "An intriguing possibility, suggested by data from experiments on *Aplysia* sensorimotor cocultures (Rajasethupathy et al., 2012), is that PSI during training obstructs DNA methylation required for memory consolidation. To test this possibility, we performed experiments in which the DNA methyltransferase (DNMT) inhibitor RG108 was injected into animals just before the onset of 5X training." As emphasized in the original review, neither the results of Rajasethupathy and colleagues nor the interpretation those authors presented suggested that protein synthesis is upstream of DNA methylation. It would be misleading to reference their paper in this context.

1b) The concept that DNA methylation provides a persistent molecular memory mechanism that survives posttraining inhibition of protein synthesis is exciting and is effectively tested by the authors' new experiments. However, none of their experiments directly tested whether PSI during training affects DNA methylation. Without direct data, whether the early protein synthesis (during training) and the DNA methylation are upstream or downstream of one another or whether they function in parallel is not clear. While there are arguments to suggest that immediate protein synthesis is upstream and later (15<minutes) synthesis is downstream, it is important to clearly acknowledge and state this existing ambiguity.

2) "Our results indicate that proteins synthesized during training (early protein synthesis) play a special, heretofore unsuspected, role in the consolidation of LTM." This is a bit of an overstatement and does not clarify exactly what is unsuspected (the priming function?). The early phase of protein synthesis has long been understood to be essential for memory consolidation. Certainly, Barzilai et al. (1989) suggested that the early wave of protein synthesis reflected expression of immediate early genes, including transcription factors that would initiate expression of genes critical for synaptic plasticity and learning. C/EBP is one such immediate early gene transcription factor in both *Aplysia* and mammals. It is important to put the priming mechanism that the authors identify into the context of immediate early gene expression. One mechanism suggested by Rajasethupathy et al. 2012 is that DNA methylation inhibits expression of CREB2, a suppressor of memory formation. With CREB2 expression reduced, initiation of the CREB-dependent transcription of immediate early genes, including transcription factors, can be enhanced. It is likely that the expression of these transcription factors represents the first wave of protein synthesis, which is sensitive to protein synthesis inhibitors during the training protocol. Rajasethupathy should be cited and this section of the manuscript clarified.

---

## [Author Response]

We thank the reviewers for their detailed, insightful comments. We have taken their criticisms seriously and, in response, have extensively rewritten the manuscript, as well as included extensive new data. In order to do this, we have converted our paper from a Research Advance into a Research Article.

*[…] Major points that must be addressed:*

*Most critical, the language and presentation are confusing and (perhaps unsurprisingly) for many of the reasons that were already brought up in the original article on which this Advance is based. This needs to be clarified, partially in the same manner as for the original article. In addition, there are some simple experimental lacunae that need to be filled out to better establish or nuance the authors' conclusions.*

*1) The term consolidation should be almost systematically replaced by an expression that conveys the main message to the reader. "Consolidation," in contemporary parlance should surely be used to include all steps from initial induction to the stable expression of memory, particularly in this simple preparation where EPSP magnitude is a satisfactory surrogate for memory expression. The long-lasting cryptic effect of 5X5HT stimulation that persists after late PSI should be termed "priming" consistent with the original manuscript. The authors' observations can be understood and communicated simply by proposing that in neurons primed by 5XHT induced early-protein synthesis, subsequent events required for LTF can be induced by 3X5HT training. This simplified presentation/ discussion would take nothing away from the novelty of the observation. Few would have thought that a long-lasting nuclear mark would persist stably in neural circuits after LTM training, if the circuit itself has never encoded stable LTM.*

*To explain our position, we point out that most neuroscientists would agree that consolidation is not one event, but involves multiple phases. One of which would be a nuclear priming event in which some nuclear processes required for LTM consolidation are activated. Another phase is revealed by this work to require late protein synthesis. There are also several other examples of multiple stages consolidation in Aplysia. The MAPK pathway is recruited relatively late during training protocols, and may well still be contributing to memory formation for tens of minutes after the last block of sensitizing stimuli (i.e., tail shocks). Chain et al. (1999) found that inhibiting PKA presynaptically within several hours after a series of 5-HT applications almost completely blocked long-term facilitation tested at 24 hours; the dependence on PKA activity appeared to gradually decrease within the first 12 hours, which presumably represents one mechanism of consolidation.*

*The overly simplistic and – we suggest – incorrect dichotomy between memory consolidation and expression, also ignores the molecular evidence that memory maintenance is an active process. The authors' previous work implicates activity of truncated PKC (PKM) in memory maintenance at times as late as 1-3 days after training.*

We completely agree with the reviewers’ reasoning and have modified the manuscript accordingly. Specifically, we no longer assert that LTM consolidation is complete by the end of training. The reviewers’ argument that the consolidation of LTM in *Aplysia* involves multiple stages over the course of several hours is well taken. As suggested, we now refer to the mnemonic component that persists after early protein synthesis as the “priming signal” or “priming component”. In addition, our results are no longer described in terms of a putative dichotomy between memory consolidation and expression. Rather, we now distinguish between a priming signal, which is induced by early protein synthesis, and consolidated LTM; the latter requires, in part, late protein synthesis

*2) On a related note, the term "retrieval" is also used in a very confusing fashion and this needs to be addressed systemically in from the Introduction through the Discussion. The authors observe that early PSI blocks EPSP potentiation (which is the mechanism of memory encoding). Early PSI does blocks memory expression/ LTF consolidation. Retrieval is never tested. The paper doesn't address the issue of memory retrieval. Rather it addresses two stages of memory encoding and the existence of a stable and rate-limiting initial process of (probably nuclear) priming.*

*In further explanation of our position, if consolidation of memory were truly complete by 30 min after training, then a protein synthesis inhibitor that blocked retrieval or expression would have no persistent effect once it had been eliminated from the CNS. If the protein synthesis inhibitor anisomycin selectively affected expression, this effect should require the presence of anisomycin at the time of the posttest. In Aplysia, the effects of anisomycin disappear within approximately 3 hours, as demonstrated by Montarolo et al. (1986). Because posttests were conducted at 24 hours and anisomycin effects should have ended at least 20 hours earlier, there should be no remaining effects on retrieval at the time of the posttest. Furthermore, the efficacy of post training anisomycin in blocking memory tested at 24 hours is highly dependent on the precise time of anisomycin administration. Montarolo et al. transiently applied anisomycin only shortly after the time point chosen in the present experiments; they found that memory retention was largely protein synthesis independent 30 min after a series of 5-HT exposures, and completely protein synthesis independent 4 hours after the end of the 5-HT. If protein synthesis inhibition impaired retrieval, the effect should be the same for applications at these several time points shortly after training. (The Montarolo et al. paper should be cited when the post training anisomycin experiment is first introduced.) Even blocking protein synthesis for 22 hours but beginning 30 min after a series of 5-HT exposures, did not dramatically block long-term facilitation measured at 24 hours. Thus, these earlier published data from Aplysia would strongly suggest that inhibition of protein synthesis 15 minutes after the end of 1.5 hour training protocol does not selectively impair expression of memory without affecting consolidation. Interestingly, although the authors cite Montarolo et al., they don't consider this earlier evidence that consolidation is still progressing at the post training time point that they apply anisomycin.*

*"Our results support a retrieval-related account of retrograde amnesia." As emphasized above, there is really no evidence presented for this conclusion.*

*"The subsequent 3X training restored LTM in the 5XTrained-Aniso-3XTrained group, as shown by the results for the 48 h posttest..… Thus, post training PSI did not block either the induction or the consolidation of LTM, but merely its expression at 24 h." This interpretation permeates the text, which leads readers, and possibly the authors, to ignore more plausible explanations for their observations, as argued above.*

We have stripped all references to retrieval from the revised manuscript. In addition, we have cited and discussed the results of Montarolo et al. (1986) immediately after the presentation of the behavioral results for posttraining anisomycin, as requested, and in the context of the presentation of the effects of posttraining anisomycin on long-term facilitation (LTF) (see subsection “LTM can be instated by truncated sensitization training following amnesia produced by posttraining PSI”, last paragraph). We now conclude that “protein synthesis during a period of 30 min or so immediately following the 5X5HT training is critical for the normal consolidation of LTF in *Aplysia*.”

*3) It is important to further assess whether test if the persistence of "priming," is reflected by the persistence of early DNA methylation, or if DNA demethylation and remethylation occur for long periods after training (as suggested by David Sweatt's 2010 analysis of fear conditioning).*

*3a) How is LTF and "priming" in response to full LTM training are affected by DNMT inhibitors applied 4 hours after training? The 4-hour time point is suggested because Montarolo and colleagues (1986) have shown after 4 hours, 24 hr LTF induced by 5X5HT treatment is independent of new protein synthesis.*

*Again, in explanation: The work here shows that whereas inhibition of protein synthesis could be "overcome" or rescued by an abbreviated additional training protocol (three bouts of tail shocks delivered at 24 hours), inhibition of DNA methyltransferases produced a memory deficit that could not be rescued by the same abbreviated additional training. Does this suggest memory maintenance requires persistent DNMT activity, as found by Miller, Sweatt and colleagues, 2010? Or is DNA methylation the signal that persists after protein synthesis inhibition (and blockade of reconsolidation) and that primes memory formation when there is additional abbreviated training? (These possibilities are not mutually exclusive.) Rajasethupathy et al. found that piRNAs are upregulated by 5-HT application during long-term facilitation, and these piRNAs promote methylation of the promoter for the inhibitor CREB, CREB2, suppressing its expression. The authors' data suggest that maintenance of DNA methylation does not require protein synthesis, but rather is perhaps upstream from protein synthesis, which is consistent with the Rajasethupathy results. Nevertheless, the authors suggest that perhaps "PSI during training obstructs DNA methylation required for memory consolidation," citing Rajasethupathy et al. This proposal should be revised, as it contradicts the authors' data and also the Rajasethupathy et al. findings. Importantly, the Rajasethupathy et al. results are consistent with the understanding that memory formation is regulated by a balance between repressors and activators of memory consolidation; the "priming effect" on memory formation by subsequent abbreviated training protocol observed here could involve suppression of expression of CREB2 due to DNA methylation, a possibility that the authors should discuss.*

In response to the reviewers’ queries regarding the potential role of DNA methylation in the consolidation and maintenance of LTM in *Aplysia*, we have added a new set of experiments in which we apply a DNMT inhibitor at 24 h and 5 d after the original (5X) sensitization experiment (Figure 6 and Figure 8). The revised manuscript also includes experiments to control for nonspecific effects of the DNMT inhibitor (RG108) used (Figure 6—figure supplement 1); to ensure that the disruptive effect of DNA methylation cannot be attributed to reconsolidation blockade (Figure 7); and to show that the disruptive effect of DNMT inhibition on LTM does not preclude the reestablishment of LTM by the full (5X) training (Figure 9). The new results demonstrate that inhibition of DNA methylation abolishes LTM—as indicated by the inability of truncated training to instate/reinstate LTM—regardless of when it is carried out. These results demonstrate a striking, unambiguous dependence of both the consolidation and maintenance of LTM on DNA methylation. As we conclude, our results imply that the consolidation of LTM requires the initial silencing of one or more genes that encode memory repressors; moreover, the maintenance of LTM also depends on the persistent silencing of this gene/these genes.

*3b) Similarly, if ongoing DNA methylation provides the molecular component of memory/priming that persists after protein synthesis inhibition, or after blockade of reconsolidation, one would predict that late treatment with a DNMT inhibitor, e.g. at 19 or 20 hours, might block the efficacy of abbreviated training in producing full LTM detectable at 48 hours. The authors should discuss this point.*

*The authors should also present additional results with RG108, to include in Figure 4, indicating how DNMT inhibition of "primed" preparations shortly before abbreviated training at the 24-hour time point affects the ability of the abbreviated training to restore LTM. Is there a reinforcement of a pre-set DNA methylation pattern required for LTM restoration? If LTM restoration is independent of ongoing DNA methylation, then the authors would have their strongest evidence for DNA methylation being a mechanism for priming. If ongoing DNA methylation is required LTM restoration, this would also be very interesting.*

As we point out (Discussion, seventh paragraph), a unique feature of the present results is that we can now make a direct comparison, for memory of one specific form of learning, of the disruption of LTM produced by reconsolidation blockade (Cai et al., 2012), inhibition of PKM (Cai et al., 2011), and inhibition of DNA methylation (present results). We show that of the three forms of mnemonic impairment, only inhibition of DNA methylation results in the inability to reinstate memory by truncated (3X) training (see Chen et al., 2014). In light of the present results, this implies that the priming signal (see our response to point 1 above) persists after reconsolidation blockade and inhibition of PKM, but is eliminated by inhibition of DNA methylation. This conclusion has important general implications for thinking about how memories are maintained.

*4) The authors should perform one experiment and discuss candidate molecular cellular processes controlled by abbreviated (3X5HT) training in naive and primed preparations. In primed preparations (24 hours after PSI delivered 15 minutes after 5X5HT training), what does protein-synthesis inhibition before and after 3X5HT stimulation do to LTM consolidation?*

*In explanation, the abbreviated training protocol with three bouts of tail shock is likely to initiate some of the same processes that contribute to memory formation as the full five-bout protocol. It is important to acknowledge that the abbreviated protocol may be sufficient for formation of memory because some residual components of memory persist after treatment with the protein synthesis inhibitor, such as DNA methylation and to provide at least one or two simple experiments to restrict the possible range of processes recruited by 3X5HT training.*

We agree with the reviewers that the molecular processes regulated by the 3X training is an important issue and one that we are starting to investigate. We respectfully request, however, that we be allowed to publish without having to experimentally address this particular issue. We believe that such experiments are outside the scope of the paper and we hope the reviewers will agree.

[Editors' note: further revisions were requested prior to acceptance, as described below.]

*Essential revisions:*

*1) Is the recruitment of DNA methyltransferases that contribute to long-term memory consolidation and maintenance downstream of protein synthesis? This is an unresolved issue, and its discussion in the manuscript needs to be improved and clarified.*

*1a) The authors state: "An intriguing possibility, suggested by data from experiments on Aplysia sensorimotor cocultures (Rajasethupathy et al., 2012), is that PSI during training obstructs DNA methylation required for memory consolidation. To test this possibility, we performed experiments in which the DNA methyltransferase (DNMT) inhibitor RG108 was injected into animals just before the onset of 5X training." As emphasized in the original review, neither the results of Rajasethupathy and colleagues nor the interpretation those authors presented suggested that protein synthesis is upstream of DNA methylation. It would be misleading to reference their paper in this context.*

*1b) The concept that DNA methylation provides a persistent molecular memory mechanism that survives posttraining inhibition of protein synthesis is exciting and is effectively tested by the authors' new experiments. However, none of their experiments directly tested whether PSI during training affects DNA methylation. Without direct data, whether the early protein synthesis (during training) and the DNA methylation are upstream or downstream of one another or whether they function in parallel is not clear. While there are arguments to suggest that immediate protein synthesis is upstream and later (15<minutes) synthesis is downstream, it is important to clearly acknowledge and state this existing ambiguity.*

We have removed the reference to the study of Rajasethupathy et al. (2012) from the Introduction to the experiments presented in Figure 4. Furthermore, in the Discussion we acknowledge that we do not have any direct evidence concerning the relationship between protein synthesis and DNA methylation, and explicitly state that protein synthesis could be either upstream or downstream of DNA methylation. See Discussion, fifth paragraph.

*2) "Our results indicate that proteins synthesized during training (early protein synthesis) play a special, heretofore unsuspected, role in the consolidation of LTM." This is a bit of an overstatement and does not clarify exactly what is unsuspected (the priming function?). The early phase of protein synthesis has long been understood to be essential for memory consolidation. Certainly, Barzilai et al. (1989) suggested that the early wave of protein synthesis reflected expression of immediate early genes, including transcription factors that would initiate expression of genes critical for synaptic plasticity and learning. C/EBP is one such immediate early gene transcription factor in both Aplysia and mammals. It is important to put the priming mechanism that the authors identify into the context of immediate early gene expression. One mechanism suggested by Rajasethupathy et al. 2012 is that DNA methylation inhibits expression of CREB2, a suppressor of memory formation. With CREB2 expression reduced, initiation of the CREB-dependent transcription of immediate early genes, including transcription factors, can be enhanced. It is likely that the expression of these transcription factors represents the first wave of protein synthesis, which is sensitive to protein synthesis inhibitors during the training protocol. Rajasethupathy should be cited and this section of the manuscript clarified.*

We are grateful to the reviewer for these insightful comments, and we have modified the revised manuscript to accommodate his critique. Specifically, in the revised manuscript we discuss the possible role of immediate-early genes in the priming component of memory, and we cite the study by Barzilai et al. (1989) as support for the idea that memory consolidation in *Aplysia* depends on early protein synthesis. Moreover, we specifically name C/EPB as one of the immediate-early genes whose expression is involved in this early protein synthesis. Finally, we point out that CREB2 is a likely candidate for the target of the DNA methylation that our results show is a critical component of memory consolidation, and cite the study by Rajasethupathy et al. (2012). We believe these changes have significantly enhanced the sophistication of our discussion of the possible mechanisms of early protein synthesis and DNA methylation in the paper. See Discussion, first paragraph.